# A PALB2-interacting domain in RNF168 couples homologous recombination to DNA break-induced chromatin ubiquitylation

Martijn S Luijsterburg[1†], Dimitris Typas[1†‡], Marie-Christine Caron[2,3], Wouter W Wiegant[1], Diana van den Heuvel[1], Rick A Boonen[1], Anthony M Couturier[2,3], Leon H Mullenders[1], Jean-Yves Masson[2,3], Haico van Attikum[1*]

[1]Department of Human Genetics, Leiden University Medical Center, Leiden, The Netherlands; [2]Genome Stability Laboratory, CHU de Québec Research Center, HDQ Pavilion, Oncology Axis, McMahon, Québec City, Canada; [3]Department of Molecular Biology, Medical Biochemistry and Pathology, Laval University Cancer Research Center, Québec City, Canada

*For correspondence: h.van. attikum@lumc.nl

[†]These authors contributed equally to this work

Present address: [‡]Ubiquitin Signaling Group, Protein Signaling Program, The Novo Nordisk Foundation Center for Protein Research, Faculty of Health and Medical Sciences, University of Copenhagen, Copenhagen, Denmark

Competing interests: The authors declare that no competing interests exist.

**Abstract** DNA double-strand breaks (DSB) elicit a ubiquitylation cascade that controls DNA repair pathway choice. This cascade involves the ubiquitylation of histone H2A by the RNF168 ligase and the subsequent recruitment of RIF1, which suppresses homologous recombination (HR) in G1 cells. The RIF1-dependent suppression is relieved in S/G2 cells, allowing PALB2-driven HR to occur. With the inhibitory impact of RIF1 relieved, it remains unclear how RNF168-induced ubiquitylation influences HR. Here, we uncover that RNF168 links the HR machinery to H2A ubiquitylation in S/G2 cells. We show that PALB2 indirectly recognizes histone ubiquitylation by physically associating with ubiquitin-bound RNF168. This direct interaction is mediated by the newly identified PALB2-interacting domain (PID) in RNF168 and the WD40 domain in PALB2, and drives DNA repair by facilitating the assembly of PALB2-containing HR complexes at DSBs. Our findings demonstrate that RNF168 couples PALB2-dependent HR to H2A ubiquitylation to promote DNA repair and preserve genome integrity.

## Introduction

Chromosomal DNA double-strand breaks (DSB) that arise in S phase during DNA replication or early in G2 phase are removed by the homologous recombination (HR) machinery, which utilizes the intact genetic information from the sister chromatid as a template for repair. Following their detection, the ends of a DSB are resected to generate extended 3' single-stranded DNA (ssDNA) overhangs, which are bound by the ssDNA-binding protein RPA. One of the key players in HR is the partner and localizer of breast cancer susceptibility protein 2 (PALB2), which is mutated in a subset of breast cancer and Fanconi Anaemia patients (FANCN) (*Xia et al., 2006*; *Reid et al., 2007*; *Antoniou et al., 2014*). PALB2 forms oligomers to interact with the breast cancer susceptibility proteins 1 and 2 (BRCA1 and BRCA2) and the recombinase RAD51 (*Sy et al., 2009*; *Zhang et al., 2009a*, *2009b*; *Buisson and Masson, 2012*). The interaction between BRCA1 and PALB2 is tightly regulated in a cell-cycle dependent manner to ensure suppression of HR in G1 (*Orthwein et al., 2015*) and activation of HR in S/G2 (*Buisson et al., 2017*). Once recruited to resected DSB ends by BRCA1, PALB2 facilitates the assembly of BRCA2 and RAD51 onto broken DNA ends, the latter of which catalyzes strand

invasion and DNA transfer during HR (*Sy et al., 2009*; *Zhang et al., 2009a*, *2009b*). This mechanism is responsible for the error-free repair of DSBs in S/G2 phase.

The alternative pathway for DSB repair is non-homologous end-joining (NHEJ), which is dominant in G1 and late G2 phase (*Karanam et al., 2012*). The choice between HR and NHEJ during the cell cycle is regulated by a signaling pathway that involves the progressive modification of chromatin surrounding DSBs (*Polo and Jackson, 2011*). This pathway involves the concerted action of the E3 ubiquitin ligases RNF8 and RNF168, which promote the ubiquitin-dependent recruitment of the BRCA1-Abraxas-RAP80-MERIT40 (BRCA1-A) and 53BP1 complexes (*Mailand et al., 2007*; *Escribano-Díaz et al., 2013*; *Stewart et al., 2009*; *Doil et al., 2009*; *Huen et al., 2007*; *Wang and Elledge, 2007*; *Kolas et al., 2007*). It has been demonstrated that the RNF168-dependent recruitment of 53BP1, through its effectors RIF1 and MAD2L2, inhibits DNA end-resection and subsequently HR (*Escribano-Díaz et al., 2013*; *Boersma et al., 2015*). However, the action of RNF168 and 53BP1 does not seem to be intrinsically inhibitory to HR. For instance, the recruitment of RIF1 and its inhibitory impact on HR are restricted to G1 cells, which favors repair by NHEJ in that cell cycle stage (*Escribano-Díaz et al., 2013*).

The accumulation of RIF1 is antagonized by BRCA1 in S/G2 phase (*Chapman et al., 2013*; *Escribano-Díaz et al., 2013*; *Feng et al., 2013*), alleviating the inhibitory impact on HR and allowing this repair pathway to reach maximal activation during these stages of the cell cycle (*Karanam et al., 2012*). With the inhibitory impact of RIF1 relieved, RNF168 is still capable of ubiquitylating chromatin in S/G2 phase (*Doil et al., 2009*), raising the question whether RNF168-induced ubiquitylation affects HR during these cell cycle stages. In this study, we reveal an unexpected role of RNF168 in physically coupling the HR machinery to H2A ubiquitylation in S/G2 cells through a newly identified PALB2-interacting domain (PID). These findings delineate a new RNF168-dependent mechanism that is paramount to PALB2 recruitment and efficient execution of HR in S/G2 phase.

## Results

### RNF168 promotes PALB2 recruitment to chromatin

To explore potential links between RNF168 and the homologous recombination (HR) machinery, we used a chromatin-tethering approach, exploiting the strong interaction between the LacR protein and the LacO DNA sequence (*Acs et al., 2011*; *Luijsterburg et al., 2012*). Human U2OS cells with stably incorporated LacO arrays (*Janicki et al., 2004*) were transfected with LacR-tagged RNF168 (*Figure 1A*), which triggered the accumulation of ubiquitin conjugates (FK2) and the ubiquitin-binding protein 53BP1 (*Figure 1—figure supplement 1A,B*) at the LacO array as previously reported (*Acs et al., 2011*; *Luijsterburg et al., 2012*; *Panier et al., 2012*). Surprisingly, we found that tethering of LacR-RNF168 also attracted PALB2 and RAD51 to the array, while LacR alone failed to do so (*Figure 1B–E*).

The RNF8 pathway has previously been implicated in the recruitment of PALB2 by promoting the accrual of the BRCA1-Abraxas-RAP80-MERIT40 (BRCA1-A) complex to DSBs (*Zhang et al., 2012*). Conversely, it was demonstrated in several other studies that RAP80 and Abraxas inhibit PALB2-dependent HR (*Coleman and Greenberg, 2011*; *Hu et al., 2011*; *Kakarougkas et al., 2013*). We recently reported that PALB2 and RAP80 reside in biochemically distinct and mutually exclusive BRCA1-containing complexes (*Typas et al., 2015*). In line with this observation, we found that RAP80 depletion did not affect PALB2 recruitment after RNF168 tethering (*Figure 1F,G*). Importantly, immunofluorescence and western blot analysis confirmed the efficient knock-down of RAP80 under these conditions (*Figure 1—figure supplement 1C,D*). Additionally, knock-down of BRCA1, which was confirmed by immunofluorescence analysis (*Figure 1H* and *Figure 1—figure supplement 1E*), also did not change the RNF168-dependent assembly of PALB2 in this experimental system (*Figure 1H,I*), suggesting that RNF168 can promote PALB2 accrual independently or downstream of BRCA1.

It should be noted that while RNF168-mediated recruitment of PALB2 in the tethering system does not require BRCA1 (*Figure 1H,I*), its recruitment into ionizing radiation-induced foci (IRIF) that contain DSBs is fully dependent on BRCA1 (*Figure 2A*), which is in agreement with earlier reports (*Zhang et al., 2009a*, *2009b*). We conclude that forced chromatin binding of RNF168 triggers PALB2 and RAD51 loading onto chromatin in a manner that does not require the RAP80-containing BRCA1-A complex.

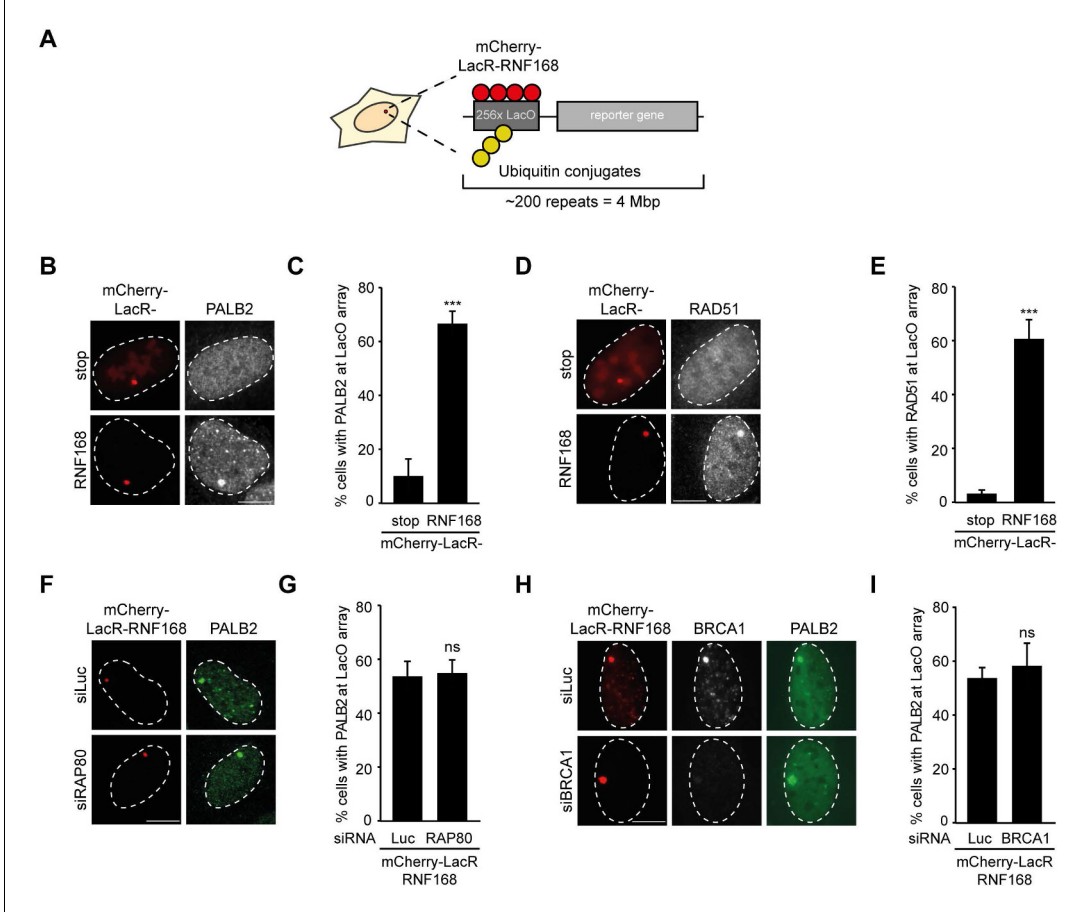

**Figure 1.** RNF168-dependent recruitment of PALB2 to chromatin. (**A**) Schematic of the tethering system. (**B**) PALB2 (white) accumulation upon tethering of the indicated mCherry-LacR fusion proteins (red) in cells containing a LacO array. (**C**) Quantification of B. (**D**) As in B, but stained for RAD51 (white). (**E**) Quantification of D. Indicated significance in C and E is compared to mCherry-LacR-stop. (**F**) As in B, but stained for PALB2 (green) in control (siLuc) or RAP80-depleted cells. (**G**) Quantification of F. (**H**) As in F, but stained for BRCA1 (white) in YFP-PALB2-expressing control (siLuc) or BRCA1-depleted cells. (**I**) Quantification of H. Indicated significance in G and I is compared to siLuc. Quantified data represent the mean ± S.E.M. (n = 3). Scale bar = 5 μm.

The following figure supplement is available for figure 1:

**Figure supplement 1.** RNF168-dependent recruitment of 53BP1 to chromatin.

## RNF168 promotes PALB2 recruitment to DSBs and execution of HR

To examine the physiological relevance of our findings in the tethering system, we decided to examine the impact of RNF8 and RNF168 on HR. It should be noted that a number of studies have previously reported a role for RNF8 in regulating HR through different mechanisms (*Huang et al., 2009*; *Lu et al., 2012*; *Zhang et al., 2012*), while other studies have reported that loss of RNF8 or RNF168 did not affect HR (*Meerang et al., 2011*; *Sy et al., 2011*; *Muñoz et al., 2012*; *Nakada et al., 2012*). Thus, the function of RNF8 and particularly RNF168 in HR is currently unclear. To elucidate the role of these proteins in this repair process, we depleted cells of RNF8 or RNF168 and monitored the recruitment of HR factors into IRIF. Cells stably expressing the mAG-tagged cell cycle marker geminin were engineered to specifically monitor recruitment in S and G2 cells (*Typas et al., 2015*). As expected, mAG-geminin-expressing cells showed recruitment of endogenous PALB2 into foci after exposure to ionizing radiation (IR) (*Figure 2A*). Importantly, a quantitative analysis of IRIF formation revealed that knock-down of RNF8 or RNF168 largely abolished the accrual of PALB2 in mAG-geminin-positive cells (*Figure 2A,D*). The knock-down of either ubiquitin ligase was verified by

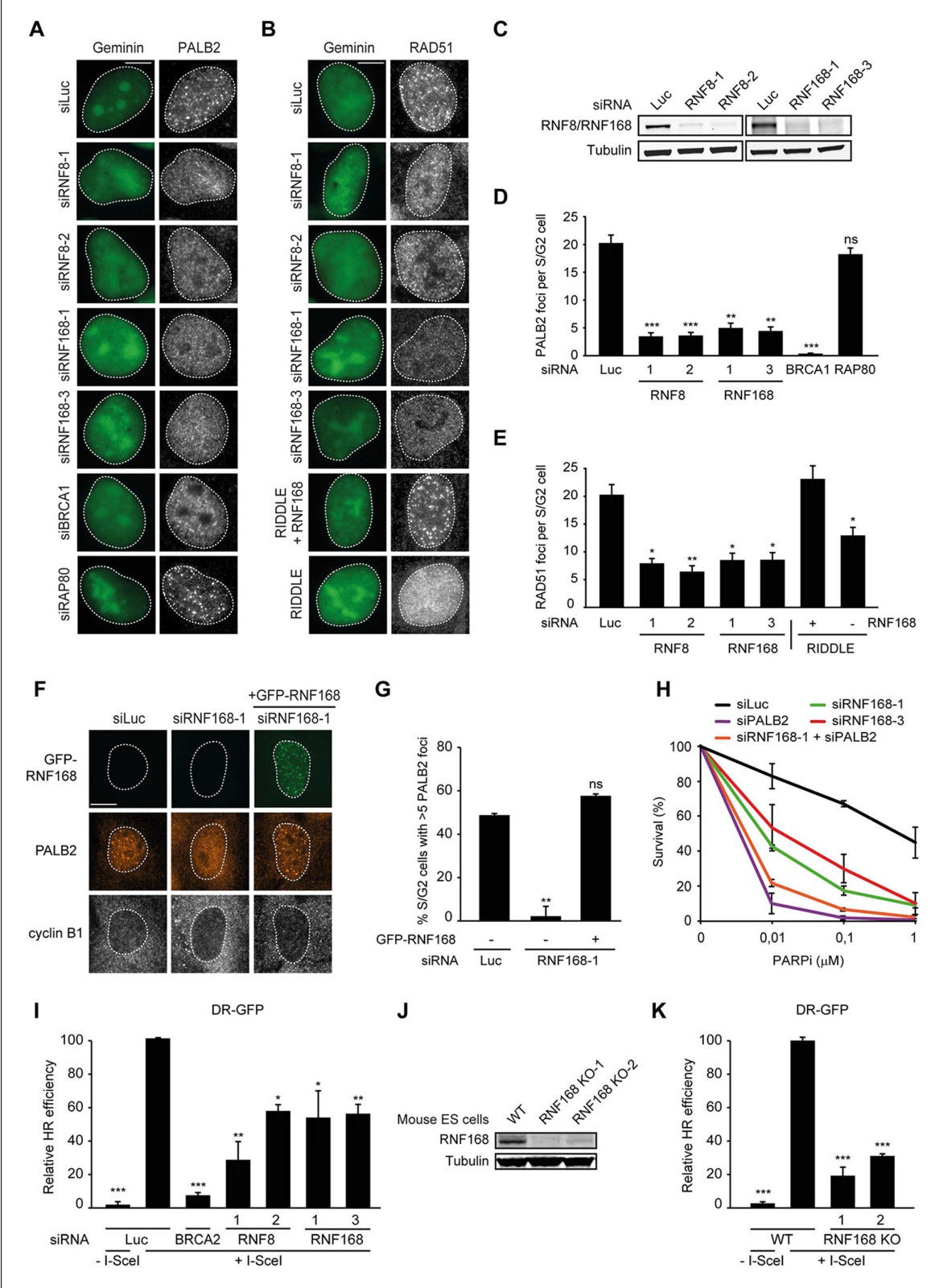

**Figure 2.** RNF168 promotes PALB2 and RAD51 recruitment and HR. (**A**) Effect of the indicated siRNAs on PALB2 (white) IRIF formation at 6 hr after 10 Gy in mAG-geminin-expressing (green) S/G2 U2OS cells. (**B**) RAD51 (white) IRIF formation at 6 hr after 10 Gy in mAG-geminin-expressing (green) S/G2 U2OS cells transfected with the indicated siRNAs and in RNF168-deficient RIDDLE S/G2 cells complemented with either an empty vector or wild-type RNF168 expression vector. (**C**) Western blot analysis of RNF8 and RNF168 expression in U2OS cells treated with the indicated siRNAs. (**D**) Quantification

*Figure 2 continued on next page*

Figure 2 continued

of A. (E) Quantification of B. Indicated significance in D and E is compared to siLuc. (F) PALB2 IRIF formation (orange) in cyclin B1-positive S/G2 U2OS cells (white) transfected with the indicated siRNAs and siRNA-resistant GFP-tagged RNF168 cDNA (green). (G) Quantification of F. Indicated significance is compared to siLuc. (H) Effect of the indicated siRNAs on the survival of U2OS cells following treatment with PARP inhibitor (PARPi) KU-0058948. (I) Effect of the indicated siRNAs on HR efficiency measured using the DR-GFP reporter in human HEK293T cells. Indicated significance is compared to siLuc + I-SceI. (J) Western blot analysis of RNF168 expression in RNF168 mouse ES knock-out clones. (K) Effect of RNF168 knock-out on HR in two individual mouse ES clones with an integrated DR-GFP reporter. Indicated significance is compared to WT + I-SceI. Quantified data are represented as mean ± S.E.M. (n = 3), except in D where n = 2. Scale bar = 5 μm.

The following figure supplement is available for figure 2:

**Figure supplement 1.** RNF168 promotes RAD51 recruitment and is epistatic with PALB2 in HR.

immunoblotting using specific antibodies (*Figure 2C*). Importantly, while BRCA1 depletion completely abolished PALB2 recruitment (*Zhang et al., 2009a*, *2009b*), loss of RAP80 had no appreciable impact (*Figure 2A,D* and *Figure 2—figure supplement 1A,B*). These results suggest that RNF8/RNF168 contribute to the BRCA1-dependent loading of PALB2 at DSBs independently of RAP80, which is in accordance with our tethering results (*Figure 1F,G*), and consistent with several studies showing that RAP80 does not contribute to, or may even antagonize HR (*Coleman and Greenberg, 2011*; *Hu et al., 2011*; *Typas et al., 2015*). Importantly, depletion of RNF8 or RNF168 also impaired efficient RAD51 assembly in cycling cells, following either IR-inflicted DNA damage (*Figure 2B,E*) or UV-A laser micro-irradiation (*Figure 2—figure supplement 1C,D*). This is in line with the effect on PALB2 recruitment, which is required for RAD51 loading (*Sy et al., 2009*; *Zhang et al., 2009a*, *2009b*). We confirmed these findings in an siRNA-independent manner by utilizing fibroblasts from an RNF168-deficient RIDDLE patient who suffers from radiosensitivity and immunodeficiency, and displays dysmorphic features (*Stewart et al., 2007*). Analysis of mAG-geminin-expressing (S/G2) RIDDLE cells revealed markedly reduced RAD51 recruitment after IR when compared to the same cells complemented with RNF168 (*Stewart et al., 2009*) (*Figure 2B,E*). To corroborate these results, we depleted endogenous RNF168 and transfected cells with siRNA-resistant GFP-tagged RNF168. Importantly, the expression of RNF168 fully rescued the loss of PALB2 foci after IR in cyclin B1-positive (S/G2) cells, demonstrating that this phenotype is not due to an off-target effect of the siRNA (*Figure 2F,G*). To further validate these findings, we assessed whether knock-down of RNF168 would render cells sensitive to PARP inhibition, which is a feature of HR-deficient cells (*McCabe et al., 2006*). RNF168-depleted cells were hypersensitive to treatment with PARP inhibitor (PARPi), although not to the extent observed after PALB2 knockdown cells (*Figure 2H*), suggesting a role for RNF168 in HR. Interestingly, the additional knock-down of PALB2 further sensitized RNF168-depleted cells to the same levels as PALB2 single knock-down cells (*Figure 2H*; *Figure 2—figure supplement 1F*), indicating that RNF168 is epistatic to PALB2 in HR. To establish whether RNF168 promotes HR, we turned to the well-established DR-GFP reporter for HR activity (*Pierce et al., 1999*). Control experiments revealed that knock-down of BRCA2 almost fully impaired the HR-mediated repair of a nuclease-induced DSB (*Figure 2I*). Using this reporter, we found that depletion of either RNF8 or RNF168 caused a significant reduction in HR efficiency (*Figure 2I*) without any impact on cell cycle distribution (*Figure 2—figure supplement 1E*). Finally, we generated RNF168 knock-out mouse embryonic stem (ES) cells using CRISPR-Cas9-mediated genome editing. The genetic ablation of RNF168 was confirmed in two independent clones by western blotting (*Figure 2J*). Both RNF168 knock-out clones showed a significant reduction in HR efficiency measured by the DR-GFP reporter (*Figure 2K*), confirming our data in human cells. Together our results suggest that RNF168 facilitates PALB2/RAD51 loading and efficient execution of HR.

## BRCA1 recruitment to HR sites does not require RNF168

The recruitment of PALB2 during HR requires both RNF168 (*Figure 2*) and BRCA1 (*Sy et al., 2009*; *Zhang et al., 2009a*, *2009b*). This raises the question whether RNF168 promotes HR at the level of PALB2 itself, or whether RNF168 acts upstream of PALB2 at the level of BRCA1. To address this, we specifically monitored BRCA1 foci formation in geminin-positive S/G2 cells at late time points post IR (6 hr after 10 Gy), when functional HR complexes have been assembled. While control cells

showed clear accumulation of BRCA1 and 53BP1 in S/G2 cells, knock-down of RNF168 still allowed BRCA1 recruitment in cells that failed to form 53BP1 foci (*Figure 3A–D*). It has previously been established that proteins involved in DSB signaling show extensive spreading from DSB sites, while proteins involved in HR show a more limited spreading, which is likely confined to resected DNA (*Bekker-Jensen et al., 2006*; *Hu et al., 2011*; *Typas et al., 2015*). As a consequence of this difference in the spreading from DSBs, signaling proteins form significantly larger IR-induced foci when compared to HR factors (see *Figure 3—figure supplement 1* for a graphical representation of this phenomenon). To confirm and extend these findings, we performed quantitative measurements on IR-induced foci size. Our analysis confirmed that signaling proteins, such as MDC1, RNF168, BRCA1 and RAP80 form foci in the order of 0.15 $\mu m^2$ in size, whereas HR proteins, such as RPA, PALB2 and RAD51 form considerably smaller foci in the range of 0.06 $\mu m^2$ (*Figure 3E,F*). Interestingly, the knock-down of RNF168 shifted the average BRCA1 foci size from that of the typical large foci (0.16 $\mu m^2$) to that in the range observed for HR proteins (0.08 $\mu m^2$; *Figure 3G*). A similar phenotype has been observed in RAP80-depleted cells both by us and by other groups (*Hu et al., 2011*; *Typas et al., 2015*). Although RNF168-depleted cells do form IR-induced BRCA1 foci in S/G2, we noted that the absolute number of BRCA1 foci is decreased compared to control S/G2 cells (*Figure 3C*). However, a direct comparison between BRCA1 foci numbers in these conditions is difficult, because RNF168-depleted cells only form small HR foci, while control cells show small HR foci in addition to large signaling foci (*Figure 3G*). In summary, our findings suggest that while RNF168 loss affects the formation of large BRCA1 foci, which arise from RAP80-dependent BRCA1 recruitment during DSB signaling, it does not impair the formation of small BRCA1 foci in S/G2 cells that are typical for factors involved in HR. This conclusion also fits with the observation that the RNF168-mediated recruitment of PALB2 in the tethering system does not require BRCA1 (*Figure 1H,I*), and suggests that RNF168 promotes HR downstream of BRCA1, likely at the level of PALB2.

## PALB2 interacts with RNF168 via its WD40 domain

Given that RNF168 could promote HR at the level of PALB2 (*Figures 2* and *3*), we sought to address if these two proteins interact with each other. To test this, we performed a pulldown assay from cell extracts that revealed an interaction between GFP-tagged RNF168 and endogenous PALB2 (*Figure 4A*). To rule out an indirect interaction between these proteins, we purified GST-tagged PALB2 and His-tagged RNF168 and addressed whether these proteins interact in vitro. In accordance with our pulldown results, we found that recombinant PALB2 efficiently bound recombinant RNF168, demonstrating a direct protein-protein interaction between these proteins (*Figure 4B*). We next set out to map the region in PALB2 that associates with RNF168. To this end, we tested a series of recombinant PALB2 deletion mutants for their interaction with RNF168 in in vitro binding assays (*Figure 4C*). Regions encompassing the ChAM domain (T3) and the C-terminal WD40 domain (T5) of PALB2 most strongly associated with RNF168 (*Figure 4C,D*). To verify whether these PALB2 domains are required for the association with RNF168 in living cells, we monitored the interaction between RNF168 and internal deletion mutants of PALB2 lacking the ChAM (T6) or WD40 domain (T7). Both PALB2$^{WT}$ and PALB2$^{\Delta ChAM}$ (T6) interacted equally strongly with RNF168, while PALB2$^{\Delta WD40}$ (T7) showed a markedly reduced ability to interact (*Figure 4E*). To reciprocally verify this result, we pulled down YFP-tagged PALB2$^{WT}$ or PALB2$^{\Delta WD40}$ (T7) and examined their association with RNF168. While RNF168 readily bound YFP-PALB2$^{WT}$, this interaction was nearly lost in the absence of the WD40 domain (*Figure 4F*). Similarly, the interaction between PALB2 and RAD51 was lost when the WD40 domain was lacking, agreeing with earlier reports (*Zhang et al., 2009a*; *Buisson et al., 2010*). The C-terminal WD40 domain is also the region in PALB2 that interacts with BRCA2 (*Oliver et al., 2009*). Indeed, we confirmed in pulldown experiments that BRCA2 readily associates with YFP-PALB2 (*Figure 4—figure supplement 1A*). We therefore asked whether RNF168, perhaps through its interaction with PALB2, can also associate with BRCA2. In addition to PALB2, we indeed found BRCA2 to also associate with GFP-RNF168 in pulldown experiments (*Figure 4G* and *Figure 4—figure supplement 1A*). To validate these findings, we performed in vitro binding assays using purified PALB2, BRCA2 and RNF168 proteins. In these experiments, we initially used a smaller chimeric BRCA2 protein of 1009 amino acids (piccolo BRCA2 or piBRCA2), which interacts with both PALB2 and RAD51 (*Buisson et al., 2010*). Indeed, we found that recombinant PALB2 readily interacted with either piBRCA2 or RNF168. When all three proteins were incubated together, we also detected an efficient association of PALB2 with both piBRCA2 and RNF168

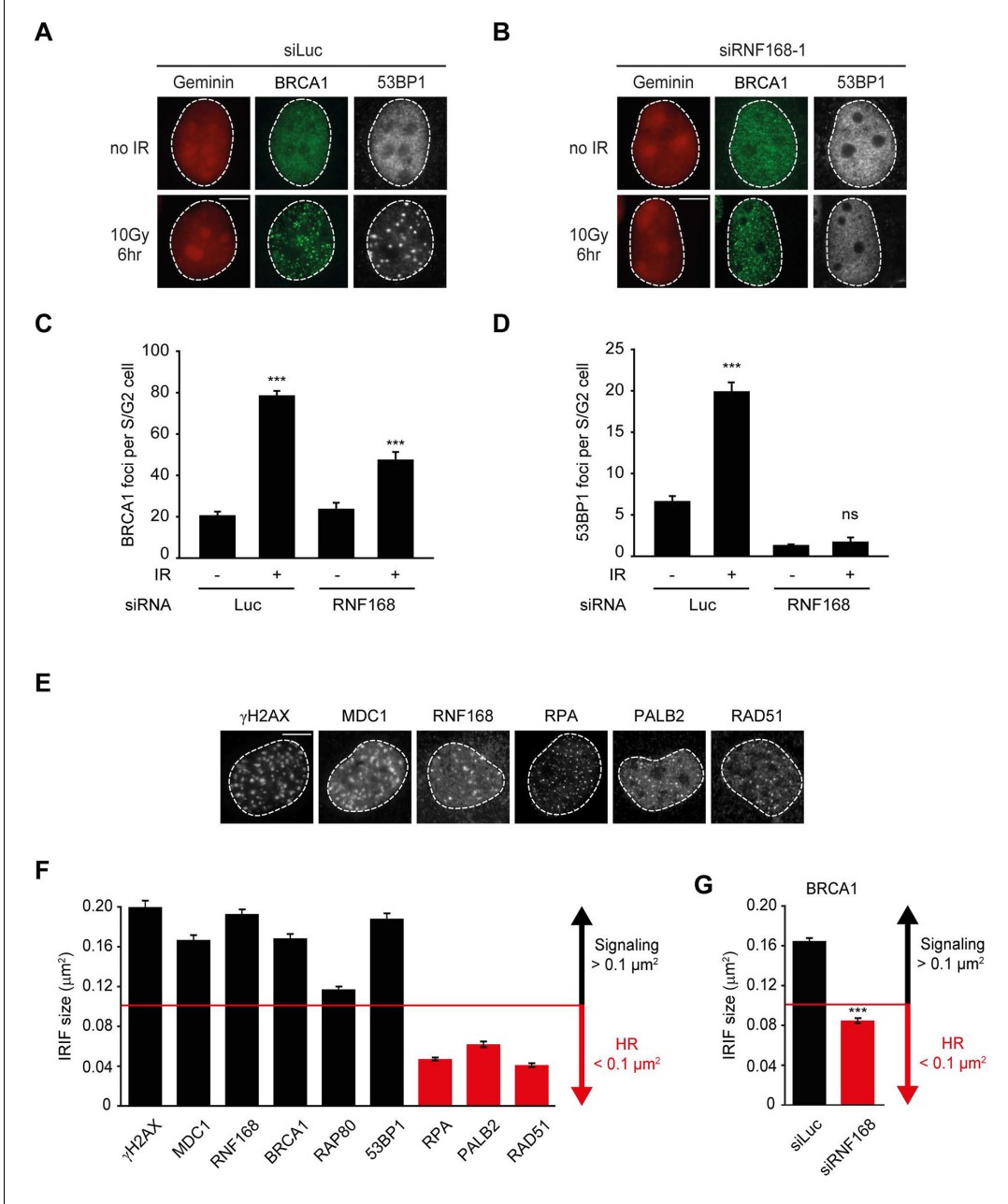

**Figure 3.** RNF168 does not affect recruitment of BRCA1 to HR sites in S/G2 cells. (**A**) BRCA1 (green) and 53BP1 (white) IRIF formation in mCherry-geminin-expressing (red) S/G2 U2OS cells transfected with siLuc. (**B**) As in A, except that cells were transfected with siRNF168. (**C**) Quantification of A. (**D**) Quantification of B. Indicated significance in C and D is compared to non-irradiated cells transfected with the same siRNA. (**E**) IRIF formation of the indicated proteins. (**F**) Quantified IRIF size (in μm²) of various signaling (black) and HR (red) proteins. (**G**) Quantified IRIF size (in μm²) of BRCA1 in U2OS cells transfected with siLuc or siRNF168. Indicated significance is compared to siLuc. Quantified data are represented as mean ± S.E.M. (n = 3). Scale bar = 5 μm.

The following figure supplement is available for figure 3:

**Figure supplement 1.** Model for the spreading of proteins from DSBs.

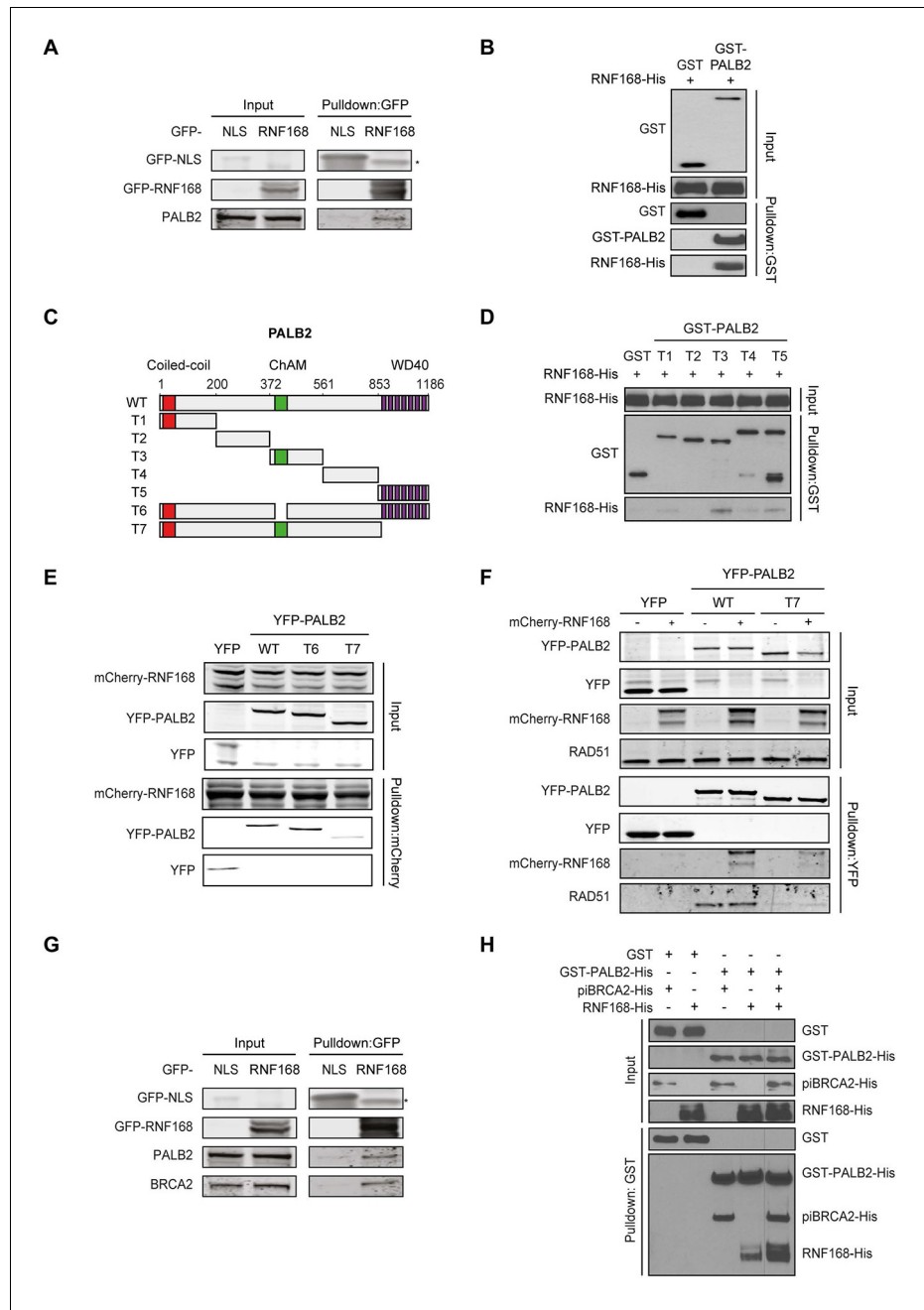

**Figure 4.** PALB2 directly interacts with RNF168 via its WD40 domain. (**A**) Pulldowns of the indicated GFP fusion proteins in U2OS cells. Blots were probed for PALB2 and GFP. (**B**) In vitro GST pulldown of GST-PALB2 or GST alone in the presence of His-RNF168. Blots were probed for GST and His. (**C**) Schematic representation of full-length PALB2 (WT), five non-overlapping fragments (T1–T5) spanning PALB2, and two deletion mutants lacking the ChAM (T6) or WD40 domain (T7). (**D**) As in B, but using the indicated GST-tagged PALB2 fragments. (**E**) mCherry-RNF168 pulldowns from U2OS cells expressing the indicated YFP- tagged fusion proteins. Blots were probed for mCherry and YFP. (**F**) Pulldowns of the indicated YFP fusion proteins in U2OS cells either with or without expression of mCherry-RNF168. Blots were probed for GFP, mCherry and RAD51. (**G**) Pulldowns of the indicated GFP fusion proteins in U2OS cells. Blots were probed for PALB2, BRCA2 and GFP. (**H**) In vitro GST pulldown of GST-PALB2 or GST alone in the presence of His-RNF168 and/or His-piBRCA2. Blots were probed for GST and His.

The following figure supplement is available for figure 4:

**Figure supplement 1.** RNF168 and BRCA2 interact with PALB2 in a non-mutually exclusive manner.

(*Figure 4H*). Similar results were obtained when using purified full-length BRCA2 protein of 3330 amino acids (*Figure 4—figure supplement 1B*). These findings show that PALB2 interacts with RNF168 through its WD40 domain in a manner that is non-mutually exclusive with BRCA2 association.

## PALB2 recruitment to DNA damage requires its WD40 domain

We next addressed whether the WD40 domain of PALB2 is required for its recruitment to chromatin (*Figure 5A*). This would be predicted considering that RNF168 facilitates the recruitment of PALB2 (*Figure 2*), and because PALB2 interacts with RNF168 through its WD40 domain (*Figure 4*). Using the chromatin-tethering system, we found that, similarly to endogenous PALB2, YFP-PALB2$^{WT}$ was recruited to chromatin following tethering of RNF168 (*Figure 5B–D*). In stark contrast, YFP-PALB2$^{\Delta WD40}$ (T7) failed to be recruited following tethering of RNF168, suggesting that the accrual of PALB2 in this system is WD40 dependent (*Figure 5C,D*). Subsequently, we assessed the recruitment of YFP-tagged PALB2 to sites of DNA damage. Importantly, we found that YFP-PALB2$^{\Delta WD40}$ (T7) did not accumulate as readily as YFP-PALB2$^{WT}$ at laser-induced DNA damage (*Figure 5E,F*), as well as at nuclease-induced DSBs (*Figure 5G–I*). To further establish the importance of the WD40 domain for PALB2 recruitment, we complemented PALB2-deficient FANCN cells with YFP-tagged PALB2$^{WT}$ or PALB2$^{\Delta WD40}$ (T7). While YFP-PALB2$^{WT}$ efficiently localized to sites of IR-induced DNA damage in cycling cells, its $\Delta$WD40 counterpart showed significantly reduced recruitment (*Figure 5J,K*). Importantly, BRCA1 was clearly recruited to nuclease-induced DSB (*Figure 5H,I*), as well as to sites of IR-induced DSBs (*Figure 5J,K*) in cells that did not show PALB2$^{\Delta WD40}$ (T7) recruitment, suggesting that the inability of PALB2$^{\Delta WD40}$ to accumulate at DSBs is not related to a defect in BRCA1 recruitment. This is in line with the observation that the interaction between BRCA1 and PALB2 relies on the N-terminal coiled-coiled domain and not the C-terminal WD40 domain of PALB2 (*Zhang et al., 2009a*, *2009b*). Together, our findings suggest that RNF168 recruits PALB2 to DSBs through its WD40 domain.

## Ubiquitin-dependent recruitment of PALB2 to DNA damage

Having shown that RNF168 facilitates PALB2 recruitment in the tethering system (*Figure 1* and *Figure 5C,D*) and at IR-induced DSBs (*Figure 2*), and having established that these proteins interact through PALB2's WD40 domain (*Figure 4*), we reasoned that RNF168 may recruit PALB2 to DSBs through a direct protein-protein interaction. Interestingly, efficient recruitment of RNF168 itself to DSBs requires an auto-amplification loop that involves the binding of RNF168 to its own ubiquitylation product through its ubiquitin-dependent DSB-recruitment module 2 (UDM2) (*Panier et al., 2012*; *Thorslund et al., 2015*). We therefore first asked whether this auto-amplification loop contributes to PALB2 accrual. To this end, we tethered catalytically inactive RNF168$^{C16S}$ to a LacO array, which resulted in substantially reduced ubiquitin conjugation compared to tethering RNF168$^{WT}$ (*Figure 6A,B*). Strikingly, catalytically dead RNF168$^{C16S}$ failed to recruit PALB2 to the LacO array in most cells (*Figure 6A,B*), with PALB2 assembly being exclusively observed in cells retaining residual ubiquitin conjugates at the array, suggesting this is due to leakiness of the RNF168$^{C16S}$ mutant (*Figure 6B*). These data suggest that PALB2 recruitment is strictly dependent on the catalytic activity of RNF168.

One explanation for these results is that PALB2 can be co-recruited to RNF168-mediated ubiquitin conjugates on DSB-neighboring chromatin through its direct interaction with RNF168. If this hypothesis is correct then PALB2 should be able to indirectly associate with ubiquitin and such an interaction should require its WD40 domain, which mediates its association with RNF168 (*Figure 4C–F*). In agreement with this rationale, recombinant GST-ubiquitin efficiently pulled down both RNF168 and YFP-PALB2$^{WT}$ from cell extracts (*Figure 6C–E*). As a control we included ubiquitin with a mutation in one of the key residues of the hydrophobic patch (I44), which inhibits the interaction with many ubiquitin-binding domains (*Hoeller et al., 2006*). Indeed, both GST alone and GST-ubiquitin$^{I44A}$ failed to pulldown either PALB2 or RNF168, showing that the hydrophobic patch on the surface of ubiquitin is required for the interaction with RNF168/PALB2. Strikingly, YFP-PALB2$^{\Delta WD40}$ (T7) failed to interact with GST-ubiquitin, indicating that the PALB2-ubiquitin interaction requires an intact WD40 domain (*Figure 6C–E*), and suggesting that RNF168 constitutes a bridge between ubiquitin and PALB2. To exclude that PALB2 itself is not capable of binding to ubiquitin,

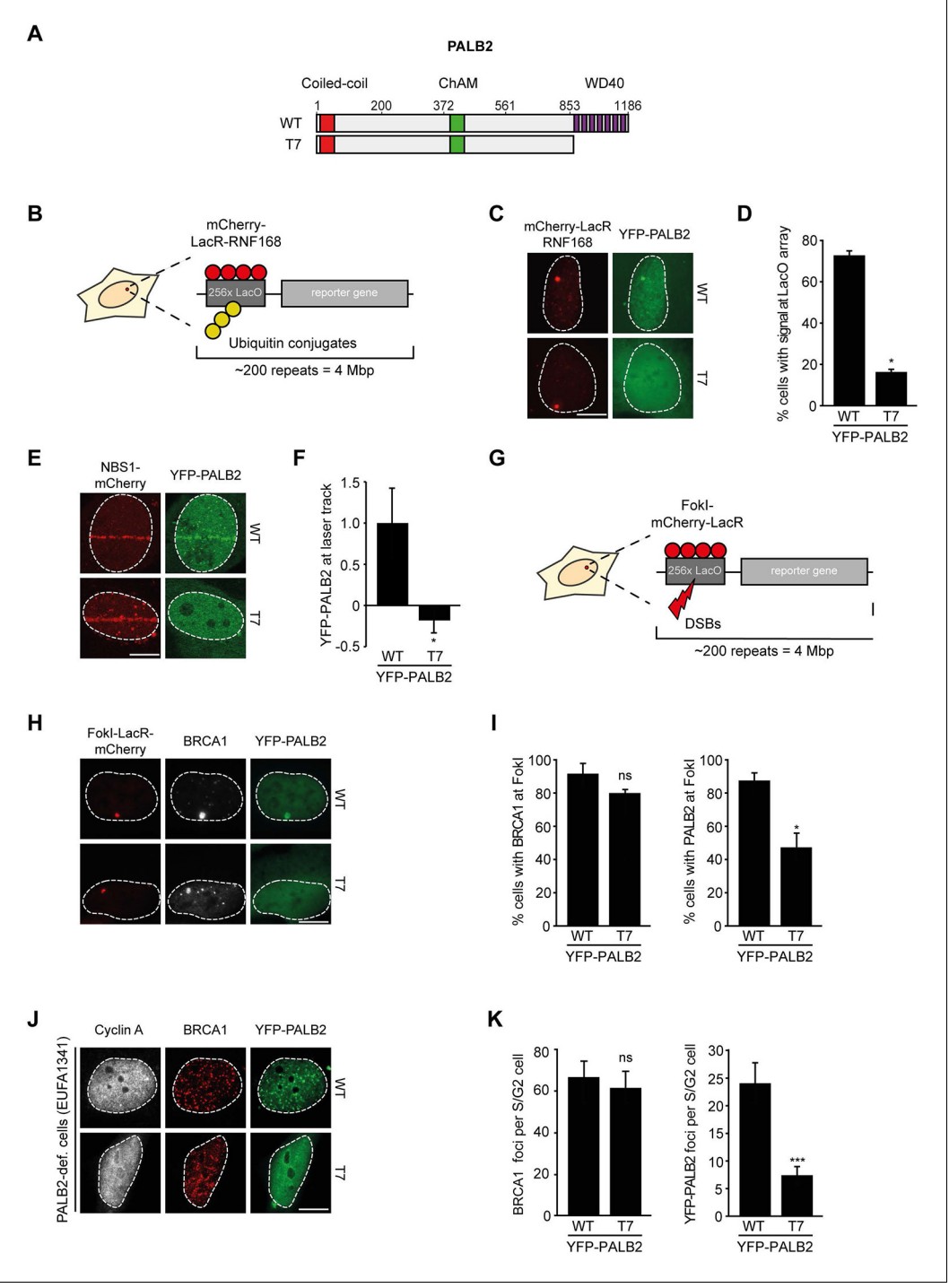

**Figure 5.** RNF168-dependent recruitment of PALB2 requires its WD40 domain. (**A**) Schematic representation of full-length PALB2 (WT) and the WD40 deletion mutant (T7). (**B**) Schematic of the tethering system. (**C**) Recruitment of the indicated YFP-PALB2 variants (green) upon tethering of mCherry-LacR-RNF168 (red) to a genomic LacO array in U2OS 2-6-3 cells. (**D**) Quantification of C. (**E**) Recruitment of the indicated YFP fusion proteins (green) to mCherry-NBS1-marked DNA damage sites (red) after multiphoton micro-irradiation in U2OS cells. (**F**) Quantification of E. (**G**) Schematic of the system used to locally induce DSBs by tethering LacR-tagged FokI endonuclease to a LacO array. (**H**) Recruitment of the indicated YFP fusion proteins (green) to BRCA1-marked (white) DSBs induced by FokI-mCherry-LacR at a LacO array (red) in U2OS 2-6-5 cells. (**I**) Quantification of H. (**J**) IRIF formation of the indicated YFP fusion proteins (green) in complemented PALB2-deficient cells (EUFA1341). S/G2 cells are marked by cyclin A (white) and BRCA1 (red) is used as a DNA damage marker. (**K**) Quantification of J.

*Figure 5 continued on next page*

*Figure 5 continued*

Indicated significance in D, F, I and K is compared to WT. Quantified data are represented as mean ± S.E.M. (n = 3), except in D and F where n = 2. Scale bar = 5 μm.

we performed in vitro interaction experiments using recombinant ubiquitin and either full-length or fragments of recombinant PALB2. Indeed, the WD40 domain of PALB2 was unable to bind poly-ubiquitin chains of various linkages under our assay conditions (K48, K63 and K27; *Figure 6F*). Moreover, none of the five PALB2 fragments spanning the entire protein, nor the full-length protein itself, was able to interact with recombinant ubiquitin (*Figure 6—figure supplement 1A,B*). These findings suggest that the WD40-dependent association of PALB2 with ubiquitin is not mediated by a direct interaction, but likely involves its ubiquitin-binding partner RNF168.

## PALB2 interacts with ubiquitin through RNF168

We next sought to address if PALB2 associates with ubiquitin conjugates through its ubiquitin-binding partner RNF168 (*Doil et al., 2009*; *Stewart et al., 2009*; *Panier et al., 2012*; *Thorslund et al., 2015*). To this end, we performed in vitro GST-binding assays using recombinant K63-linked poly-ubiquitin chains, as these are the chains preferentially bound by RNF168 (*Pinato et al., 2011*). As expected, we found that purified RNF168 could interact with K63-polyubiquitin chains, but this association was not stimulated in the presence of PALB2 (*Figure 6—figure supplement 1C*). In line with our previous data, we found that purified PALB2 was unable to bind K63-polyubiquitin chains on its own, but it was proficient in doing so when recombinant RNF168 was included in the reaction (*Figure 7A*). Thus, our in vitro reconstitution experiments reveal an RNF168-dependent association of PALB2 with poly-ubiquitin conjugates.

## PALB2 recruitment requires H2A ubiquitylation at K13/K15

It was demonstrated that RNF168 specifically ubiquitylates the core histone H2A at N-terminally situated lysines 13 and 15 in response to DNA damage (*Mattiroli et al., 2012*; *Fradet-Turcotte et al., 2013*). To address if these histone marks contribute to PALB2 recruitment, we made use of a point-mutant of RNF168 (R57D), which is still able to ubiquitylate, but has no activity toward H2A in a nucleosomal context (*Mattiroli et al., 2012*). In contrast to RNF168$^{WT}$, tethering LacR-tagged RNF168$^{R57D}$ failed to recruit 53BP1 and PALB2, suggesting that RNF168-mediated H2A ubiquitylation promotes PALB2 recruitment (*Figure 7B,C*). To corroborate these findings, we used a replacement strategy in which cells depleted for endogenous RNF168 were transfected with siRNA-resistant versions of GFP-tagged RNF168. Ectopic expression of GFP-RNF168$^{WT}$ fully rescued both PALB2 (*Figure 7D,E*) and 53BP1 (*Figure 7—figure supplement 1A,B*) recruitment into IRIF. Conversely, neither RNF168$^{C16S}$ nor RNF168$^{R57D}$ were proficient in restoring PALB2 or 53BP1 foci formation after IR (*Figure 7D,E* and *Figure 7—figure supplement 1A,B*). Importantly, GFP-RNF168$^{R57D}$, like its wild-type counterpart, was still recruited into IRIF (*Figure 7D*) or to nuclease-induced DSBs (*Figure 7—figure supplement 1C*), albeit not as efficiently as wild-type RNF168. To further test the contribution of RNF168-mediated ubiquitylation of H2A on K13/K15, we over-expressed FLAG-tagged versions of either H2A$^{WT}$ or H2A$^{K13/15R}$ in U2OS cells harboring a stably integrated LacO array and an inducible version of FokI-mCherry-LacR. Tethering of FokI-mCherry-LacR and subsequent DSB formation at the array triggered the robust recruitment of YFP-PALB2 in FLAG-H2A$^{WT}$-expressing cells (*Figure 7F*). Conversely, expression of FLAG-H2A$^{K13/15R}$ caused a significant decrease in YFP-PALB2 recruitment even though the accumulation of FokI-mCherry-LacR was similar between cells expressing H2A$^{WT}$ or H2A$^{K13/15R}$ (*Figure 7F,G*). These results indicate a strong dominant-negative effect of the FLAG-H2A$^{K13/15R}$ mutant on PALB2 recruitment. We also witnessed a considerable decrease in 53BP1 recruitment in H2A$^{K13/15R}$-expressing cells, demonstrating the validity of this approach (*Figure 7—figure supplement 2A,B*). Together, these results show that the catalytic activity of RNF168 toward H2A on K13/15 is required to promote efficient PALB2 recruitment. We propose that the initial RING-dependent ubiquitylation of H2A by RNF168, and the subsequent MIU-dependent binding of additional RNF168 proteins to this ubiquitin mark creates a chromatin-

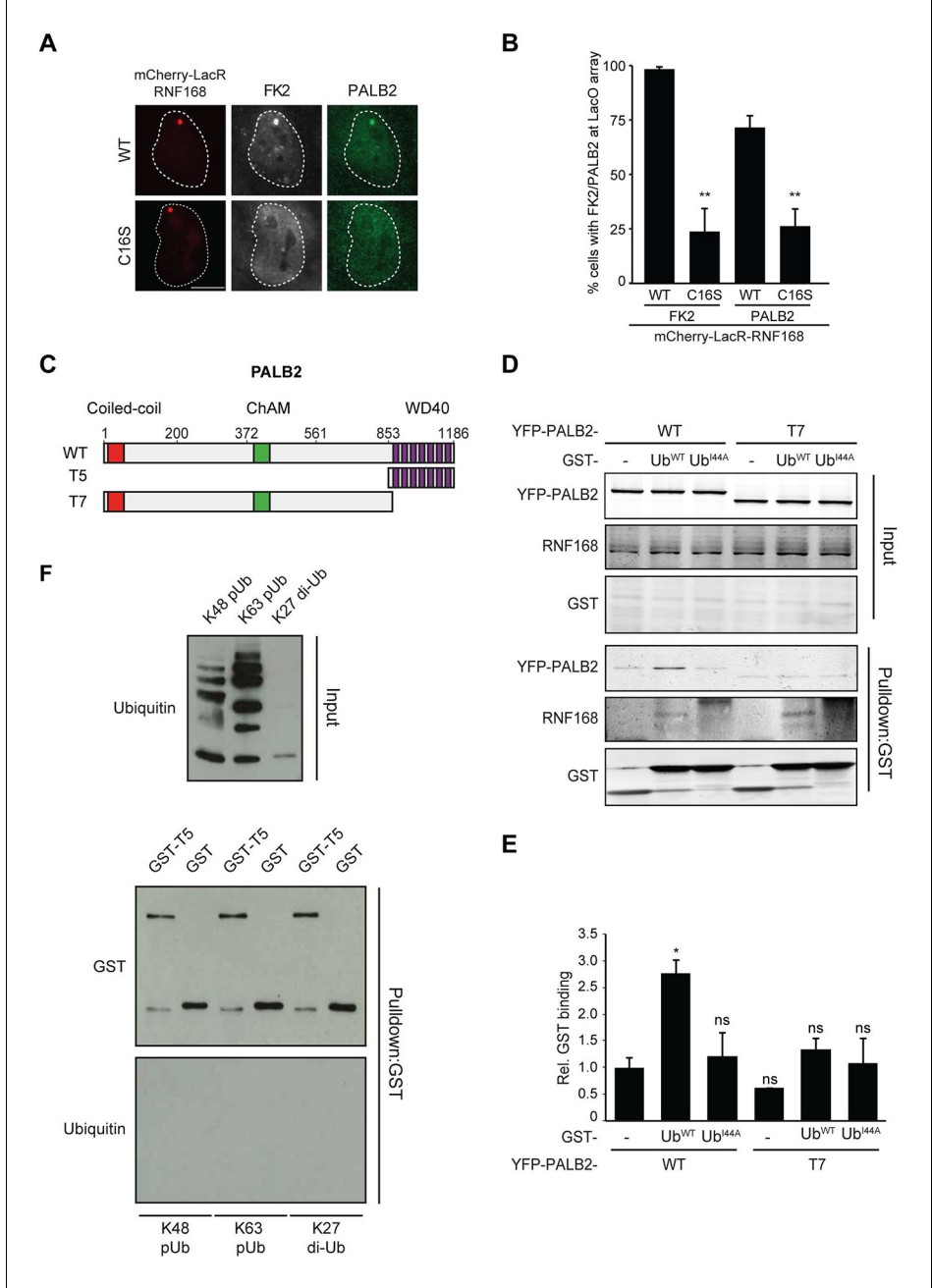

**Figure 6.** RNF168-mediated PALB2 recruitment is ubiquitylation dependent. (**A**) Recruitment of ubiquitin conjugates (FK2; white) and endogenous PALB2 (green) upon tethering of the indicated mCherry-LacR-RNF168 variants (red) to a genomic LacO array in U2OS 2-6-3 cells. (**B**) Quantification of A. Indicated significance is compared to WT for each staining. (**C**) Schematic representation of full-length PALB2 (WT), a fragment spanning the WD40 (T5) and the WD40 deletion mutant (T7). (**D**) Pulldowns of GST alone or GST-ubiquitin (WT or I44A) from U2OS cell extracts containing the indicated YFP-PALB2 fusion proteins. Blots were probed for PALB2, RNF168 and GST. (**E**) Quantification of D in which the PALB2/GST ratios of each pulldown were normalized to YFP-PALB2$^{WT}$/ GST alone. Indicated significance is compared to YFP-PALB2$^{WT}$/GST. Quantified data are represented as mean ± S.E.M. (n = 3). (**F**) Pulldown of GST only or GST-T5 in the presence of polyubiquitin (pUb) chains of K48 or K63 linkage, or di-ubiquitin (di-Ub) chains of K27 linkage. Blots were probed with GST and ubiquitin.

The following figure supplement is available for figure 6:

**Figure supplement 1.** PALB2 neither associates with ubiquitin, nor stimulates the association of RNF168 with ubiquitin in vitro.

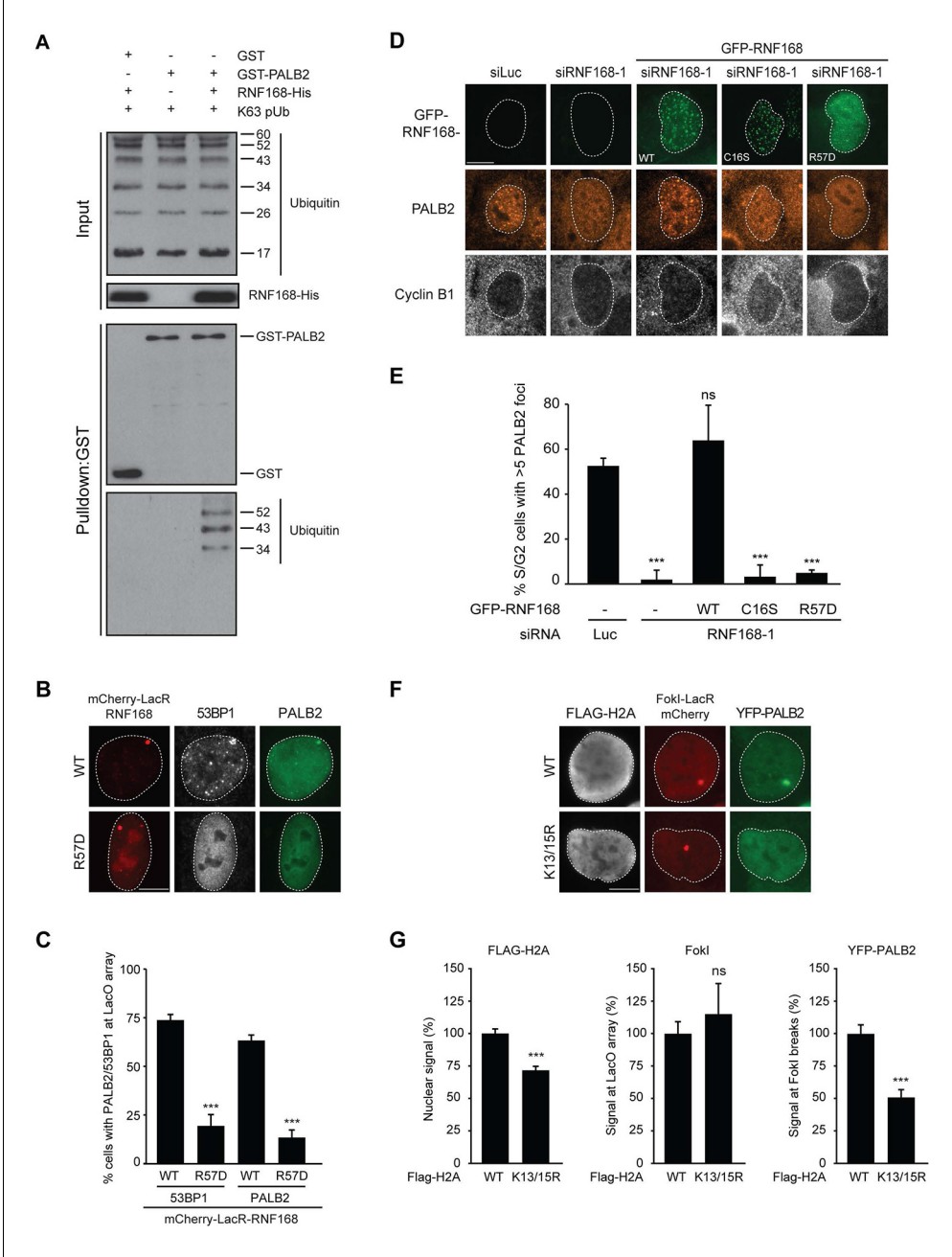

**Figure 7.** PALB2 associates with K63-polyubiquitin chains via RNF168. (**A**) GST pulldowns to assess the binding of GST-PALB2 or GST-alone to K63 polyubiquitin (pUb) chains in the presence or absence of His-RNF168. Blots were probed for GST, ubiquitin and His. (**B**) Recruitment of 53BP1 (white) and YFP-PALB2 (green) upon tethering of the indicated mCherry-LacR-RNF168 variants (red) to a genomic LacO array in U2OS 2-6-3 cells. Indicated significance is compared to WT for each staining. (**C**) Quantification of B. (**D**) PALB2 IRIF formation (orange) in cyclin B1-positive U2OS cells (white) transfected with the indicated siRNAs and siRNA-resistant GFP-tagged RNF168 cDNAs (green). (**E**) Quantification of D. Indicated significance is compared to siLuc. (**F**) Recruitment of YFP-PALB2 (green) to DSBs induced by FokI-mCherry-LacR at a LacO array (red) in U2OS 2–6–5 cells expressing the indicated FLAG-H2A constructs (white). (**G**) Quantification of F. Indicated significance is compared to WT for each staining. Quantified data are represented as mean ± S.E.M. (n = 3), except in E where n = 4. Scale bar = 5 μm.
The following figure supplements are available for figure 7:

**Figure supplement 1.** RNF168 RING mutants do not support IR-induced 53BP1 focus formation.

*Figure 7 continued*

**Figure supplement 2.** Expression of H2A lacking K13 and K15 decreases 53BP1 recruitment to FokI-induced DSBs.

**Figure supplement 3.** Model for RNF168's role in PALB2 recruitment.

bound binding platform for the recruitment of PALB2 (see *Figure 7—figure supplement 3* for a model).

## PALB2 recruitment requires its interaction with the C-terminus of RNF168

Having shown that PALB2 recruitment requires the RNF168-dependent ubiquitylation of H2A (*Figure 7*), and considering that RNF168 and PALB2 interact at the protein level (*Figure 4*), we next sought to solidify that the interaction between these proteins is a prerequisite for the efficient recruitment of PALB2 to sites of DNA damage. Already supporting this possibility was our finding that the WD40 domain in PALB2 is required for both its interaction with RNF168 (*Figure 4*), as well as its efficient recruitment to DSBs (*Figure 5*). However, considering that the WD40 domain of PALB2 also serves as the interaction surface for a number of other proteins (*Pauty et al., 2014*), we set out to map the region in RNF168 that is required for the interaction with PALB2. To this end, we generated a series of deletion mutants and tested their ability to interact with endogenous PALB2 in pulldown assays (*Figure 8A–D*). Using this approach we found that GFP-tagged RNF168$^{WT}$, RNF168$^{\Delta1-189}$ (T1) and RNF168$^{\Delta191-382}$ (T2) associated with PALB2 with equal efficiency (*Figure 8A, B*), suggesting that the N-terminus and central region of RNF168 are not required for this interaction. Conversely, RNF168$^{\Delta384-571}$ (T3) showed an almost complete loss of PALB2 binding, indicating that the C-terminus of RNF168 mediates its interaction with PALB2 (*Figure 8A–D*). Considering that RNF168$^{\Delta384-571}$ (T3) lacks the UDM2 required for the recruitment of full-length RNF168, we generated additional C-terminal deletion mutants that do include this domain (*Figure 8A*). Analysis of these mutants revealed that RNF168$^{\Delta479-571}$ (T4) and RNF168$^{\Delta525-571}$ (T5) also failed to efficiently bind PALB2 (*Figure 8A–D*). We conclude that the C-terminal ~50 amino acids of RNF168 comprise a new PALB2-interacting domain, which we have named PID (for PALB2-Interacting Domain). An alignment of the PID in human RNF168 revealed considerable conservation of this domain in mammalian species (*Figure 8—figure supplement 1*). Having identified which region in RNF168 mediates its association with PALB2, we finally examined the functional relevance of this region for PALB2 recruitment. We reconstituted RNF168-depleted cells with siRNA-resistant GFP-tagged wild-type RNF168 or C-terminal deletion mutants of RNF168 that lack the PID and are therefore unable to interact with PALB2 (*Figure 8A*). For this analysis we only used RNF168 mutants RNF168$^{\Delta479-571}$ (T4) and RNF168$^{\Delta525-571}$ (T5) since these mutants, in contrast to RNF168$^{\Delta384-571}$ (T3), contain the UDM2 and are therefore proficient in accumulating at DSBs. Expression of these GFP-RNF168 variants rescued the 53BP1 foci defect in RNF168-depleted cells (*Figure 8—figure supplement 2A–C*), which shows that these mutants are recruited to DSBs and are able to ubiquitylate H2A. Conversely, while expression of RNF168$^{WT}$ fully restored PALB2 foci formation in geminin-positive S/G2 cells, the C-terminal deletion mutants RNF168$^{\Delta479-571}$ (T4) and RNF168$^{\Delta525-571}$ (T5) failed to fully rescue this PALB2 foci defect (*Figure 8E,F*). These data further prove that the RNF168-mediated recruitment of PALB2 depends on a direct protein-protein interaction that is mediated by the C-terminal PID of RNF168. To further corroborate these findings, we assessed the relevance of the PID of RNF168 for PALB2-driven HR in PARPi survival assays. To this end, we used the Flp-In/T-REx system to establish HeLa cells stably expressing inducible siRNA-resistant GFP-tagged RNF168 alleles that were either able (WT) or unable (T5) to interact with PALB2. HeLa cells stably expressing GFP-NLS were generated to serve as a control. To perform PARPi survival assays, we knocked-down endogenous RNF168 with siRNAs in these cells and induced ectopic expression of RNF168 alleles with doxycycline (DOX). The expression was shut down by a DOX wash-out 12–24 hr before cells were subjected to a 24 hr pulse of 1 µM PARPi (*Figure 8G*), resulting in ectopic RNF168 expression levels in the near-physiological range during the treatment with PARPi (*Figure 8H*; *Figure 8—figure supplement 3*). Using this

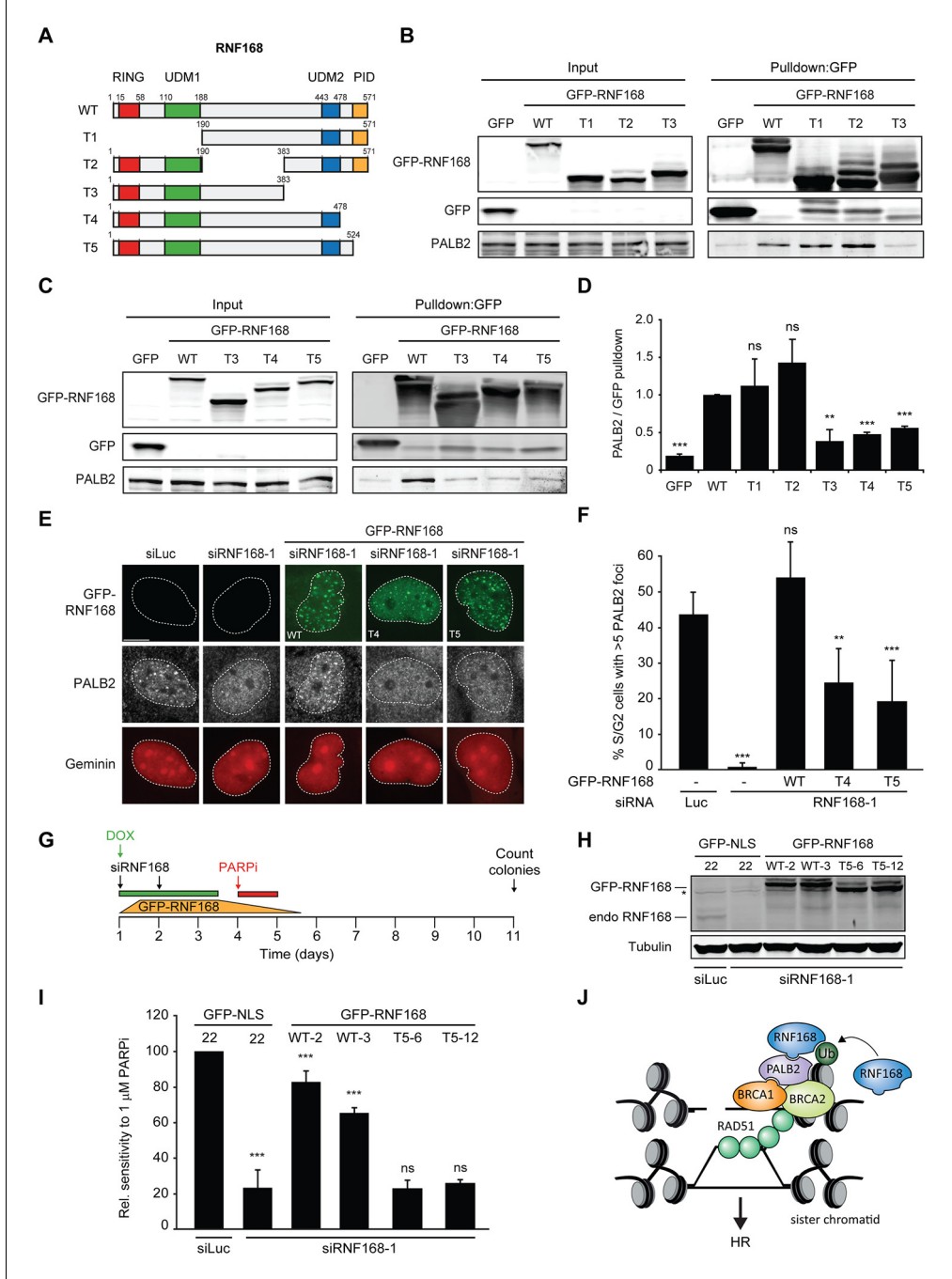

**Figure 8.** Recruitment of PALB2 requires its interaction with RNF168's PID. (**A**) Schematic representation of full-length RNF168 (WT) and five deletion mutants (T1–T5). (**B, C**) Pulldowns of the indicated GFP fusion proteins in U2OS cells. Blots were probed for PALB2 and GFP. (**D**) Quantification of B and C. Indicated significance is compared to WT. (**E**) PALB2 IRIF formation (white) in mCherry-geminin-positive U2OS cells (red) transfected with the indicated siRNAs and siRNA-resistant GFP-tagged RNF168 cDNAs (green). (**F**) Quantification of E. Indicated significance is compared to siLuc. (**G**) Schematic overview of the PARPi survival experiment using independent HeLa Flp-In/TRex clones with inducible expression of GFP-tagged RNF168 proteins. (**H**) Expression levels of endogenous RNF168 and ectopic GFP-RNF168$^{WT}$ or GFP-RNF168$^{\Delta525-571}$ (T5) at day 4 of the experiment outlined in G. Tubulin serves as a loading control. The star indicates a non-specific band. (**I**) Effect of ectopic expression of the indicated GFP-tagged proteins on the survival of stable HeLa cells after transfection with the indicated siRNAs and treatment with 1 μM PARP inhibitor (PARPi) KU-0058948 according to the experimental set-up outlined in G. Two independent HeLa clones expressing RNF168$^{WT}$ (clone 2 and 3) or RNF168$^{\Delta525-571}$(T5) (clone 6 and

*Figure 8 continued on next page*

*Figure 8 continued*

12) were used. Hela GFP-NLS clone 22 was included as a control. Quantified data are represented as mean ± S.E.M. (n = 2), except in F where n = 5. Scale bar = 5 μm. (J) Model for the trimodular recruitment of PALB2 to HR sites.

The following figure supplements are available for figure 8:

**Figure supplement 1.** Conservation of the RNF168 PID in mammals.
**Figure supplement 2.** RNF168 PID mutants fully support IR-induced 53BP1 focus formation.
**Figure supplement 3.** Stable inducible expression of GFP-tagged proteins in HeLa cells.

approach, we found expression of GFP-RNF168$^{WT}$ to almost fully rescue the PARPi sensitivity of RNF168-depleted cells, while expression of GFP-NLS or GFP-RNF168$^{\Delta525-571}$ (T5) failed to rescue this defect (*Figure 8I*). Notably, we observed this effect in two independent clones expressing RNF168$^{WT}$ or RNF168$^{\Delta525-571}$ (T5). These findings suggest that the interaction between PALB2 and the C-terminal PID of RNF168 contributes to efficient DSB repair by HR. In summary, our findings support a piggyback ride mechanism for PALB2 recruitment, which links the ability of RNF168 to amplify and bind ubiquitin conjugates at damaged chromatin to a direct protein-protein interaction between PALB2's WD40 domain and RNF168's C-terminal PID (*Figure 8J*).

## Discussion

In this study, we define a new ubiquitin-dependent mechanism that regulates efficient HR. We demonstrate that RNF168 drives the recruitment of PALB2 to DSBs in a manner depending on its ubiquitin ligase activity toward histone H2A on K13/K15. Moreover, we show that RNF168 recruits PALB2 to DSBs through a direct protein-protein interaction that is established by the newly identified C-terminal PID of RNF168 and the C-terminal WD40 domain of PALB2. These findings advance our mechanistic understanding of the RNF168-dependent response to DSBs by revealing a physical and functional link between RNF168 and the core HR protein PALB2.

### RNF168 promotes DSB repair through HR

The repair of DSBs in S/G2 cells relies on HR, which involves the sequential assembly of a functional repair complex consisting of BRCA1, PALB2, BRCA2 and RAD51. The accumulation of PALB2 depends on BRCA1 and is subsequently required for the efficient recruitment of BRCA2 and RAD51 (*Sy et al., 2009*; *Zhang et al., 2009a*, *2009b*). Our current study reveals an unexpected role of the ubiquitin ligase RNF168 in promoting HR in S/G2-phase cells. Firstly, we show that knock-down of RNF168 impairs the recruitment of PALB2 and RAD51 to DSBs in S/G2 phase. This phenotype is also observed in cells from RNF168-deficient RIDDLE patients. Moreover, re-expression of siRNA-resistant RNF168 in RNF168-depleted cells, and of RNF168 in RIDDLE cells fully restored the HR defects, showing that these are a specific consequence of impaired RNF168 protein function. Secondly, we demonstrate that knock-down of RNF168 in human cells or knock-out of RNF168 in mouse ES cells confers a defect in HR measured by the well-established DR-GFP reporter. Thirdly, we find that knock-down of RNF168 leads to pronounced PARP inhibitor sensitivity, which is an established hallmark of defective HR. Previous studies also lend support for a role of RNF168-induced H2A ubiquitylation in promoting HR-mediated DSB repair. For example, cells deficient of the USP51 deubiquitylase, or the TRIP12 and UBR5 ubiquitin ligases display elevated levels of RNF168-induced H2A ubiquitylation at DSBs, which is accompanied by an increase in HR efficiency (*Gudjonsson et al., 2012*; *Wang et al., 2016*). Our results provide a mechanistic basis for these reported findings by showing that RNF168 couples the HR machinery to H2A ubiquitylation in S/G2 cells. Although the role of RNF168, and to a lesser extent RNF8, in regulating HR has remained controversial largely due to conflicting results (*Huang et al., 2009*; *Meerang et al., 2011*; *Sy et al., 2011*; *Lu et al., 2012*; *Muñoz et al., 2012*; *Nakada et al., 2012*; *Zhang et al., 2012*), our study solidifies the importance of RNF168 in promoting efficient HR. Consistently, we find that RNF8, which

facilitates the recruitment of RNF168 to damaged chromatin through ubiquitylation of histone H1 (*Thorslund et al., 2015*), also contributes to HR. However, we cannot exclude the possibility that RNF8 promotes HR by targeting substrates other than H1 for ubiquitylation (*Huang et al., 2009*; *Lu et al., 2012*).

## Auto-amplification of RNF168 recruitment contributes to PALB2 accrual

The RNF168 protein contains at least three domains required for its function. Firstly, the UDM1 domain associates with RNF8-generated ubiquitin conjugates on histone H1, which is the initial step of ubiquitin-dependent DSB signaling (*Panier et al., 2012*; *Thorslund et al., 2015*). Secondly, once recruited to RNF8-generated ubiquitin conjugates through UDM1, RNF168 utilizes its catalytic RING domain to ubiquitylate histone H2A at K13 and K15 (*Mattiroli et al., 2012*). Thirdly, the UDM2 domain in RNF168 promotes its association with the ubiquitin conjugates on H2A. Once bound, RNF168 can further ubiquitylate neighboring H2A using its RING domain and bind to these moieties through the UDM2. This auto-amplification through the UDM2 is critical for efficient RNF168 recruitment to DSBs (*Panier et al., 2012*; *Thorslund et al., 2015*). Our findings suggest that the ability of RNF168 to promote its own recruitment is also an integral part of PALB2-dependent HR. We show that both the catalytically inactive mutant RNF168$^{C16S}$ (*Doil et al., 2009*) and the RNF168$^{R57D}$ mutant, which is not able to bind and ubiquitylate nucleosomal H2A (*Mattiroli et al., 2012*), fail to promote PALB2 recruitment. Both these mutants could neither support PALB2 recruitment in the LacR/LacO tethering system nor to DSBs in complementation experiments with RNF168-depleted cells. Furthermore, we find that over-expression of H2A$^{K13/15R}$, which lacks the ubiquitylation sites targeted by RNF168, inhibited PALB2 recruitment to nuclease-induced DSBs. Together, these findings suggest that the ubiquitin-dependent accumulation of RNF168 on broken chromosomes is critical for PALB2 recruitment.

## RNF168 recruits PALB2 through protein-protein interactions

The significance of RNF168 in PALB2 recruitment is further corroborated by in vitro studies with purified RNF168 and PALB2, which reveal a direct protein-protein interaction. Domain mapping uncovered that the C-terminally located WD40 domain of PALB2 is the key determinant for this interaction. In line with RNF168 supporting PALB2 recruitment through a direct interaction, we find that full-length PALB2 lacking its WD40 domain is recruited much less efficiently to laser-, nuclease-, and IR-induced DSBs. Interestingly, biochemical reconstitution experiments reveal that PALB2 indirectly associates with poly-ubiquitin chains through a WD40-dependent interaction with recombinant RNF168. Thus, we can fully reconstitute the interaction between PALB2, RNF168 and poly-ubiquitin, providing a molecular mechanism for the RNF168-dependent recruitment of PALB2. Domain mapping also revealed that PALB2 associates with a previously unrecognized region in the C-terminus of RNF168, which we termed the PALB2-Interacting Domain (PID). Intriguingly, RNF168 mutants lacking the PID proved to be separation-of-function mutants which fully support 53BP1 foci formation, but are defective in supporting PALB2 accumulation at DSBs. Similarly, we found that stable ectopic expression of RNF168$^{WT}$ rescued the PARPi sensitivity of RNF168-depleted cells, while a mutant lacking the PID (RNF168$^{\Delta525-571}$ (T5)) failed to do so. These findings illustrate the importance of the protein-protein interaction between PALB2 and RNF168 in HR, and unveil the presence of a previously unrecognized domain in the C-terminus of RNF168 that is critical for its role in HR. One interesting possibility is that RNF168 through this direct protein-protein interaction targets PALB2 for ubiquitylation. However, whether this is the case and whether this potential RNF168-dependent ubiquitylation of PALB2 has functional significance for HR remains to be established.

## A new RNF168-dependent mechanism for PALB2 recruitment

Based on our findings, we propose the following model for RNF168-dependent regulation of HR (*Figure 8J*): RNF168 decorates histones and other proteins with ubiquitin following its binding to RNF8-modified chromatin in the ssDNA-containing compartment proximal to DSBs. Although it may be expected that resected DNA is devoid of histones, previous studies have shown that resection can actually occur in a nucleosomal context (*Tsukuda et al., 2005*; *Costelloe et al., 2012*). RNF168 auto-amplifies its accumulation in the ssDNA compartment and recruits PALB2 through a direct protein-protein interaction. Once recruited, PALB2, in turn, facilitates BRCA2 and RAD51 assembly and

promotes efficient HR. We propose that these events contribute to a trimodular mechanism for PALB2 recruitment in which (1) BRCA1 promotes PALB2 recruitment in an RNF168-independent manner through a protein-protein interaction involving the coiled-coil domain situated at the N-terminus of PALB2 (*Zhang et al., 2009a, 2009b*), (2) the more centrally located chromatin-association motif (ChAM) of PALB2 stabilizes its binding to damaged chromatin in the ssDNA compartment, and (3) the C-terminal WD40 domain of PALB2 further facilitates its retention by associating with the C-terminal PID of RNF168. We propose that integrating these independent recruitment signals leads to the efficient and specific accrual of PALB2 to HR sites (*Figure 8J*). Such a multifaceted recruitment mechanism is not uncommon in the DSB response. For instance, the assembly of 53BP1 at damaged chromatin requires the simultaneous presence of two distinct marks on histones H4 and H2A (*Fradet-Turcotte et al., 2013*; *Tuzon et al., 2014*). It is clear that the action of RNF168 in chromatin more distal to DSBs results in the ubiquitin-dependent recruitment of 53BP1 (*Bunting et al., 2010*) and the BRCA1-Abraxas-RAP80-MERIT40 complex (*Coleman and Greenberg, 2011*; *Hu et al., 2011*), both of which are inhibitory to DNA-end resection. It is important to note that the RNF168-mediated modification of H2A at K13/15 and the subsequent association of 53BP1 to this mark are not intrinsically inhibitory to HR. The inhibition of end resection is mediated downstream of 53BP1 by the recruitment of RIF1, which is restricted to G1 cells. Thus, RNF168 is only inhibitory to HR in G1 (*Chapman et al., 2013*; *Escribano-Díaz et al., 2013*; *Orthwein et al., 2015*). Indeed, our results reveal that RNF168 promotes PALB2 accrual and subsequent RAD51 loading on resected DNA in S/G2 cells to promote HR. These findings demonstrate the versatility of the RNF168-mediated ubiquitylation of H2A in regulating HR in a cell cycle-dependent manner.

## Materials and methods

### Cell culture

All cells were grown in DMEM (Gibco) containing 10% FCS (Bodinco BV) at 37°C in 5% $CO_2$. Human U2OS and HEK293 cells were purchased from ATCC. RNF168-deficient immortalized human RIDDLE fibroblasts and their FLAG-RNF168-complemented counterparts were a kind gift of Grant Stewart. FANCN/PALB2-deficient immortalized EUFA1341 fibroblasts were a gift of Johan de Winter. U2OS 2–6–3 cells containing 200 copies of a LacO-containing cassette were a gift of Susan Janicki (*Janicki et al., 2004*). A modified version of these cells stably expressing an inducible version of ER-mCherry-LacR-FokI-DD (2-6-5 cells) were a gift of Roger Greenberg (*Tang et al., 2013*). U2OS and RIDDLE cells stably expressing mAG- or mCherry-geminin were generated by lentiviral infection. HeLa Flp-In/T-REx cells, which were generated using the Flp-In/T-REx system (Thermo Fisher Scientific), were a gift of Geert Kops and Stephen Taylor. These cells were used to stably express inducible version of GFP-NLS, GFP-RNF168$^{WT}$ or GFP-RNF168$^{\triangle525-571}$ (T5) by co-transfection of pCDNA5/FRT/TO-Puro plasmid encoding GFP or GFP-tagged RNF168 (WT or T5) proteins (5 µg), together with pOG44 plasmid encoding the Flp recombinase (0.5 µg). After selection on 1 ug/mL puromycin, single clones were isolated and expanded. Stable HeLa Flp-In/T-REx clones were incubated with 50 ng/mL doxycycline to induce expression of cDNAs. All cells were authenticated by STR profiling and tested negative in routinely performed mycoplasma tests.

### Plasmids

YFP-PALB2$^{WT}$ (encoding a protein of 1186 amino acids; ENST00000261584) and YFP-PALB2$^{\triangle ChAM}$ (T6) were previously described (*Bleuyard et al., 2012*). YFP-PALB2$^{\triangle WD40}$ (T7) was generated by inserting a fragment spanning the first 2874 nucleotides of PALB2$^{WT}$ cDNA into YFP-C1. The mCherry-LacR-stop plasmid was previously described (*Soutoglou and Misteli, 2008*). GFP-C1-RNF168 was a gift of Jiri Lukas. RNF168-His plasmid was a gift of Niels Mailand. FLAG-H2A$^{WT}$ and FLAG-H2A$^{K13/15R}$ plasmids were a gift of Lorenza Penengo (*Gatti et al., 2012*). GFP-RNF168$^{\triangle1-189}$ (T1) and RNF168$^{\triangle191-382}$ (T2) were a gift of Daniel Durocher (*Panier et al., 2012*). GST-PALB2 fragments T1-T5 were previously described (*Buisson et al., 2010*). The LacR cDNA was inserted into mCherry-C1 to generate mCherry-LacR-C1. The RNF168 cDNA containing inactivating point mutations in both MIU's (A179G, A450G) was inserted into this vector to generate mCherry-LacR-RNF168$^{\triangle MIU}$. Site-directed mutagenesis was used to introduce C16S (catalytically dead) or R57D (unable to associate with H2A) in this background. An siRNF168–1-resistant version of GFP-

RNF168$^{WT}$ (a gift of Titia Sixma) or RNF168$^{R57D}$ was generated by introducing the underlined mutations C ACA CTA AGT CCA CAA ATC as previously described (*Mattiroli et al., 2012*). Moreover, PCR was used to generate siRNA-resistant GFP-RNF168$^{\triangle 384-571}$ (T3), RNF168$^{\triangle 479-571}$ (T4) and RNF168$^{\triangle 525-571}$ (T5). Fragments encoding these GFP-RNF168 constructs were subsequently inserted into pcDNA5/FRT/TO-Puro. NBS1-mCherry was previously described (*Luijsterburg et al., 2016*).

## Transfections

Plasmid transfections were performed using Lipofectamine 2000. Cells were transfected with plasmid DNA using Lipofectamine 2000 (Invitrogen) or JetPEI (Polyplus) according to the manufacturer's instructions. Cells were typically analyzed 24 hr after transfection. All siRNA transfections (see *Supplementary file 1* for siRNA sequences) were performed with 40 nM siRNA duplexes using Lipofectamine RNAiMAX (Invitrogen). Cells were transfected once or twice with siRNAs at 0 and 24 hr and were typically analyzed 60 hr after the first transfection.

## Generation of DSBs

IR was delivered by a YXlon X-ray generator (YXlon International, 200 KV, 10 mA, dose rate 2 Gy/min).

## FokI assays

U2OS 2-6-3 cells expressing inducible ER-mCherry-LacR-FokI-DD (a gift of Roger Greenberg) (*Tang et al., 2013*) were treated with 300 nM 4-OHT (Sigma) and 1 µM Shield-I (Clontech) for 5 hr. Subsequently, cells were fixed with formaldehyde and immunostained with the indicated antibodies.

## Multiphoton laser micro-irradiation

U2OS cells grown on 18 mm coverslips were placed in a Chamlide CMB magnetic chamber and the growth medium was replaced by $CO_2$-independent Leibovitz's L15 medium supplemented with 10% FCS and penicillin-streptomycin (Invitrogen). Laser micro-irradiation was carried out on a Leica SP5 confocal microscope equipped with an environmental chamber set to 37°C. DSB-containing tracks (1.5 µm width) were generated with a Mira modelocked titanium-sapphire (Ti:Sapphire) laser (l = 800 nm, pulse length = 200 fs, repetition rate = 76 MHz, output power = 80 mW) using a UV-transmitting 63 × 1.4 NA oil immersion objective (HCX PL APO; Leica). Confocal images were recorded before and after laser irradiation at 5 or 10 s time intervals over a period of 2–3 min.

## UV-A laser micro-irradiation

U2OS cells were grown on 18 mm coverslips and sensitized with 10 µM 5′-bromo-2-deoxyuridine (BrdU) for 24 hr as described (*Luijsterburg et al., 2016*). For micro-irradiation, the cells were placed in a Chamlide TC-A live-cell imaging chamber that was mounted on the stage of a Leica DM IRBE widefield microscope stand (Leica) integrated with a pulsed nitrogen laser (Micropoint Ablation Laser System; Andor). The pulsed nitrogen laser (16 Hz, 364 nm) was directly coupled to the epifluorescence path of the microscope and focused through a Leica 40x HCX PLAN APO 1.25–0.75 oil-immersion objective. The growth medium was replaced by $CO_2$-independent Leibovitz's L15 medium supplemented with 10% FCS and penicillin-streptomycin and cells were kept at 37°C. The laser output power was set to 72 to generate strictly localized sub-nuclear DNA damage. Following micro-irradiation, cells were incubated for the indicated time-points at 37°C in Leibovitz's L15 and subsequently fixed with 4% formaldehyde before immunostaining. Typically, an average of 50 cells was micro-irradiated (two iterations per pixel) within 10–15 min using Andor IQ software (Andor).

## Microscopy analysis

Images were acquired on a Zeiss AxioImager D2 widefield fluorescence microscope equipped with 40x, 63x and 100x PLAN APO (1.4 NA) oil-immersion objectives (Zeiss) and an HXP 120 metal-halide lamp used for excitation. Images were recorded using ZEN 2012 software. IRIF were evaluated in ImageJ, using a custom-built macro that enabled automatic and objective analysis of the foci as described previously (*Typas et al., 2015*).

## Immunofluorescent labeling

Immunofluoresecent labeling was carried out as described previously (*Luijsterburg et al., 2012*; *Smeenk et al., 2013*). Cells were either directly fixed or pre-extracted with 0.25% Triton-X100 (Serva) in cytoskeletal (CSK) buffer (10 mM Hepes-KOH, 300 mM Sucrose, 100 mM NaCl, 3 mM MgCl2, pH 7.4) for 2 min and subsequently fixed with 4% formaldehyde in PBS for 15 min. Cells were post-extracted with 0.5% Triton-X100 (Serva) in PBS, and treated with 100 mM glycine in PBS for 10 min to block unreacted aldehyde groups. Cells were rinsed with phosphate-buffered saline and equilibrated in wash buffer (WB; PBS containing 0.5% BSA, and 0.05% Tween 20 (Sigma-Aldrich)). Antibody steps and washes were in WB. The primary antibodies (see *Supplementary file 1* for a list of antibodies) were incubated overnight at 4°C. Detection was done using goat anti-mouse or goat anti-rabbit Ig coupled to Alexa 488, 546 or 647 (1:1000; Invitrogen Molecular probes). Samples were incubated with 0.1 µg/ml DAPI and mounted in Polymount (Polysciences).

## GST-ubiquitin binding assay

Ubiquitin-binding assays were performed as described (*Hoeller et al., 2006*). HEK293T cells were transfected with the indicated YFP-tagged PALB2 constructs, lysed for 10 min on ice in lysis buffer (50 mM HEPES, 150 mM NaCl, 1 mM EDTA, 1 mM EGTA, 10% glycerol, 1% Triton-X-100, 25 mM NaF, 10 µM $ZnCl_2$, pH 7.5) containing protease and phosphatase inhibitor cocktails (Roche). Cell lysates were collected, centrifuged for 15 min (13,000g) to remove the insoluble fraction and incubated with GST, GST–Ubiquitin$^{WT}$ or GST–Ubiquitin$^{I44A}$ (UBPBio) coupled to Glutathione sepharose 4B (Amersham Biosciences) for 4 hr at 4°C. After incubation, the sepharose matrix was washed three times with lysis buffer. Bound proteins were eluted with 10 mM reduced glutathion and analysed by immunoblotting.

## Immunoprecipitation

U2OS or HEK293T cells were transfected with plasmids encoding variants of YFP-PALB2 or GFP/mCherry-RNF168. For immunoprecipitation, cells were lysed in EBC-150 buffer (50 mM Tris, pH 7.5, 150 mM NaCl, 0.5% NP-40, 1 mM EDTA) with 500 U Benzonase supplemented with protease and phosphatase inhibitor cocktails (Roche). The lysed cell suspension was incubated for 1 hr under rotation. The cleared lysates were subjected to GFP immunoprecipitation with GFP Trap beads (Chromotek). The beads were then washed 4–6 times with EBC-150 or 300 buffer (50 mM Tris, pH 7.5, 150/300 mM NaCl, 0.5% NP-40, 1 mM EDTA) and boiled in sample buffer. Bound proteins were resolved by SDS-PAGE and immunoblotted with the indicated antibodies.

## Western blotting

Cell extracts were generated by heating cell pellets in Laemmli buffer to 95°C or 55°C (for detection of BRCA2), separated by SDS-PAGE and transferred to Immobilon-FL PVDF membranes (Millipore). Membranes were probed with the antibodies listed in *Supplementary file 1*, followed by protein detection using the Odyssey infrared imaging scanning system (LI-COR Biosciences). Secondary antibodies were purchased from Biotium.

## Generation of RNF168 knock-out mouse ES cells

A modified version of DR-GFP (p59X DR-GFP) containing an inactivated puromycin-resistance marker and an upstream hygromycin-coding sequence was targeted to the *Pim1* locus in mouse IB10 ES cells (derived from 129/Ola E14). Single copy integration was verified by Southern blot analysis and by PCR. An antisense guide RNA (5'-ACTGACATTCGGCTAAGGAA-3') targeting the first exon of RNF168 was inserted into the pSpCas9(BB)−2A-GFP vector (PX458; Addgene #48138). Transfected DR-GFP mouse ES cells were sorted by flow cytometry for GFP expression, plated at low density after which individual clones were isolated. Knock-out of RNF168 in the isolated clones was verified by western blot analysis and Sanger sequencing.

## HR assay

HEK293T, or mouse ES cells (WT or RNF168-/-) containing a stably integrated copy of the DR-GFP reporter were used to measure the repair of I-SceI-induced DSBs by HR as described (*Pierce et al., 1999*). Briefly, 48 hr after siRNA transfection, the cells were co-transfected with an mCherry

expression vector and the I-SceI expression vector pCBASce. 48 hr later the percentage of GFP-positive cells among mCherry-positive cells was determined by FACS on a BD LSRII flow cytometer (BD Bioscience) using FACSDiva software version 5.0.3.

## PARP inhibitor survival

U2OS cells were transfected with siRNA on day 1 and 2. On day 4, the cells were trypsinized, seeded at low density and mock-treated or exposed to 0.01, 0.1 and 1 µM PARP inhibitor KU-0058948 (Astrazeneca). On day 11, the cells were washed with 0.9% NaCl and stained with methylene blue. Colonies of more than 20 cells were scored. Stable HeLa Flp-In/T-Rex cells with inducible versions of GFP-NLS, GFP-RNF168$^{WT}$ or GFP-RNF168$^{\triangle525-571}$ (T5) were transfected with siRNA on day 1 and 2 in the presence of 50 ng/mL doxycycline to induce expression of GFP-tagged proteins. To shut down ectopic expression of GFP-tagged proteins, doxycycline was removed by extensive washes with PBS between 12–24 hr prior to trypsinization, seeding at low density followed by mock-treatment or exposure to a 24 hr pulse of 1 µM PARPi on day 4. On day 11, the cells were washed with 0.9% NaCl and stained with methylene blue. Colonies of more than 20 cells were scored.

## Cell cycle profiling

For cell cycle analysis cells were fixed in 70% ethanol, followed by DNA staining with 50 µg/ml propidium iodide in the presence of RNase A (0.1 mg/ml) (Sigma). Cell sorting was performed on an LSRII flow cytometer (BD Bioscience) using FACSDiva software (version 5.0.3; BD). Quantifications were performed using Flowing Software.

## Protein purification

GST-PALB2-His was purified from baculovirus-infected Sf9 cells as described in *Buisson et al. (2010)* and the protein was either eluted with 25 mM Glutathione (GE Healthcare) or cleaved with PreScission protease (GE Healthcare). Recombinant PALB2 GST fusions (T1-T5) were purified from BL21 (DE3) RP (Stratagene) as described in (*Buisson et al., 2010*) and the proteins were eluted with 25 mM Glutathione. Recombinant FLAG-piBRCA2-His and BRCA2-FLAG were purified as described (*Buisson et al., 2014*). Recombinant RNF168-His protein was purified from *E. coli* BL21(DE3) RP cells (Stratagene), grown at 37°C in Luria broth medium supplemented with 100 µg/mL ampicillin and 25 µg/mL chloramphenicol. At OD600 = 0.4, 80 µg/mL ampicillin was added to the media, and at OD600 = 0.8, 0.1 mM IPTG (Sigma) was added to the culture and incubated at 15°C overnight (16 hr). The cells were harvested by centrifugation, frozen on dry ice and stored at −80°C. Cells were lysed in P5 buffer (50 mM NaHPO$_4$ pH 7.0, 500 mM NaCl, 10% glycerol, 0.05% Triton-X-100, 5 mM imidazole) containing 0.5 mM DTT and EDTA-free protease inhibitors (Roche) and homogenized by 20 passes through a Dounce homogenizer (pestle A). The cell lysate was incubated with 1 mM MgCl$_2$ and 2.5 U/ml benzonase nuclease (Millipore) at 4°C for 1 hr followed by centrifugation at 35,000 rpm for 30 min. The soluble cell lysate was subjected to Talon affinity column purification as described (*Maity et al., 2013*) and eluted with 500 mM imidazole. The purified protein was dialysed in storage buffer (20 mM Tris-Acetate pH 8.0, 200 mM KAc, 10% glycerol, 1 mM EDTA, 0.5 mM DTT) and stored at −80°C. GST-RNF168 protein was purchased from Abnova.

## In vitro ubiquitin binding by PALB2

Purified PALB2 fusions (T1-T5) (1500 ng) or GST (1500 ng) were incubated with 2 µg of purified RNF168-His, 10 µg of unlabelled ubiquitin (VIVA Bioscience) or 4 µg of the polyubiquitin chains of K48 or K63 linkage (2–7 moities, Boston Biochem.) or 1 µg of di-ubiquitin (K27 linkage, Boston Biochem.) for 15 min at 37°C in 100 µl of GSTB buffer (20 mM Hepes pH 7.4, 125 mM KCl, 0.2 mM EDTA, 10% Glycerol, 0.5 mM DTT, 0.2% NP-40 and 1 mg/ml bovine serum albumin (BSA)) containing protease inhibitors (Roche). Then 40 µl of Glutathione Sepharose beads (GE Healthcare) and 400 µl of GSTB buffer were added for 20 min at room temperature. Beads were washed four times with GSTB buffer without BSA and eluted with 40 µl of Laemmli buffer. Proteins were visualized by western blotting using the indicated antibodies in *Supplementary file 1*. Purified GST-PALB2-His (1 µg) or GST (1 µg) in presence or absence of UbiK63 (1 µg) were incubated with RNF168-His (4 µg) for 35 min at room temperature in GSTB buffer containing 150 mM KCl. Purified GST-RNF168 (500 ng) or GST (500 ng) were incubated in presence of K63 pUb (1 µg) and with or without PALB2-His (250 ng)

for 35 min at room temperature in GSTB buffer containing 150 mM KCl. The GST pulldowns were performed as described above. For GST pulldowns in the presence of piBRCA2 or full-length BRCA2, purified GST-PALB2-His or GST were incubated with RNF168-His (500 ng) for 10 min at 37°C in GSTB buffer containing 150 mM KCl followed by addition of FLAG-piBRCA2-His or BRCA2-FLAG for 10 min at 37°C. The GST pulldowns were performed as described above. For GST pulldowns in the presence of piBRCA2, the quantities of proteins used in the assay are the following: GST-PALB2-His, GST and FLAG-piBRCA2-His: 200 ng; RNF168-His: 500 ng. For GST pulldowns in the presence of BRCA2-FLAG, the quantities of proteins used are the following: GST-PALB2-His, GST and BRCA2-FLAG: 125 ng; RNF168-His: 250 ng.

## Statistical analysis

Statistical significance was assessed by a two-tailed, unpaired t-test and is indicated as: ***$p < 0.001$, **$p < 0.01$, *$p < 0.05$. ns, not significant. All error bars represent the S.E.M.

## Acknowledgements

The authors thank Drs. S Janicki, R Greenberg, G Stewart, J de Winter, A Miyawaki, M Jasin, B Xia, F Esashi, N Mailand, J Lukas, L Penengo, D Durocher, T Sixma, G Kops and S Taylor for generously providing reagents. JYM was a FRQS Chercheur National Investigator and is a FRQS research chair in genome stability. This work was supported by grants to MSL (VENI from Netherlands Organization for Scientific Research (NWO-ALW) and LUMC Research Fellowship), AMC (Luc Bélanger scholar), LHM (Netherlands Toxicogenomics Centre (NTC)), JYM (Canadian Institutes of Health Research), HvA (Dutch Cancer Society (KWF) and ERC Consolidator).

## Additional information

### Funding

| Funder | Grant reference number | Author |
|---|---|---|
| Netherlands Organisation for Scientific Research | 863.11.007 | Martijn S Luijsterburg |
| Leiden University Medical Center | | Martijn S Luijsterburg |
| Netherlands Toxicogenomics Centre | 050-060-510 | Leon H Mullenders |
| Canadian Institute of Health Research | | Jean-Yves Masson |
| Dutch Cancer Society | 2014-6843 | Haico van Attikum |
| European Research Council | 617485 | Haico van Attikum |

The funders had no role in study design, data collection and interpretation, or the decision to submit the work for publication.

### Author contributions

MSL, Conceptualization, Data curation, Formal analysis, Supervision, Funding acquisition, Validation, Investigation, Visualization, Methodology, Writing—original draft, Writing—review and editing; DT, Conceptualization, Data curation, Formal analysis, Validation, Investigation, Visualization, Methodology, Writing—original draft, Writing—review and editing; M-CC, WWW, Data curation, Formal analysis, Investigation, Visualization, Methodology; DvdH, RAB, Data curation, Formal analysis, Investigation, Visualization; AMC, Resources; LHM, Funding acquisition, Writing—review and editing; J-YM, Conceptualization, Data curation, Formal analysis, Supervision, Funding acquisition, Visualization, Writing—review and editing; HvA, Conceptualization, Resources, Supervision, Funding acquisition, Investigation, Visualization, Methodology, Writing—original draft, Project administration, Writing—review and editing

Author ORCIDs
Anthony M Couturier, http://orcid.org/0000-0002-1512-9558
Haico van Attikum, http://orcid.org/0000-0001-8590-0240

## Additional files

**Supplementary files**

• Supplementary file 1. siRNAs and antibodies used in this study. (A) List of siRNAs and (B) List of antibodies.

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
