## [Decision Letter]

Thank you for submitting your article "A PALB2-interacting domain in RNF168 couples homologous recombination to DNA break-induced chromatin ubiquitylation" for consideration by *eLife*. Your article has been reviewed by three peer reviewers, and the evaluation has been overseen by a Reviewing Editor and Jessica Tyler as the Senior Editor. The following individuals involved in review of your submission have agreed to reveal their identity: Yossi Shiloh (Reviewer #2) and Alex Sartori (Reviewer #3).

The reviewers have discussed the reviews with one another and the Reviewing Editor has drafted this decision to help you prepare a revised submission.

Summary:

Interesting study that brings novel mechanistic insights into the DNA damage response (DDR). Papers in this field are highly relevant to a wide audience in the life sciences. Luijsterburg et al. reveal how a central player in the early stage of the DSB response, the ubiquitin E3 ligase RNF168, is involved in mediating a critical DSB repair pathway – homologous recombination repair (HRR), which takes place in late S or G2. The work shows how RNF168 mediates the recruitment to DSB sites of a critical HR player – PALB2. The core finding is the physical interaction between RNF168 and PALB2, which explains how PALB2 reads histone ubiquitylation – a basic and essential PTM in this pathway. The work provides evidence that RNF168 and PALB2 directly interact and propose further that complex formation involves the C-terminus of RNF168 (referred to as the PALB2 interaction domain, PID) and the WD40 domain of PALB2.

Essential revisions:

Since it has been previously shown that BRCA2 binds to the WD40 β-propeller domain of PALB2, the fact that RNF168 interacts with the exact same region of PALB2 (Figure 4) raises a few questions that should be discussed in more detail (subsection “A new RNF168-dependent mechanism for PALB2 recruitment”); especially in the light of the simplified model shown in Figure 8 that is lacking BRCA2 and RAD51. For instance, do RNF168 and BRCA2 interact with PALB2 in a mutually exclusive manner? Does RNF168 eventually compete with BRCA2 for PALB2 binding? How is the potential handover of PALB2 from RNF168 to BRCA2 regulated?

All interaction studies were performed with (transiently) overexpressed, epitope-tagged versions of RNF168. To address this issue, you could co-immunoprecipitate endogenous RNF168 from cell lysates and probe the bound fractions for PALB2 and BRCA2. If this is not possible for technical reasons, GFP-RNF168 pulldown fractions (Figure 4) should also be probed for the presence of endogenous BRCA2.

It is shown that RNF168 depletion results in hypersensitivity to olaparib treatment and decreased HR efficiency (Figure 2), but not to the same extent as compared to BRCA2 depletion, which is expected. The question remaining is whether RNF168 acts epistatically to PALB2 with regards to HR or in genetically separable pathways. It is suggested to perform clonogenic PARP inhibitor survival assays using single and combined depletion of RNF168 and PALB2.

The PALB2-RNF168 interactions shown in Figure 8 are not convincing. The proteins are grossly overexpressed. A question remaining is whether the endogenous proteins interact.

To further corroborate the role of the RNF168-PALB2 interaction in HR, it would be important to examine olaparib hypersensitivity of RNF168-depleted cells (Figure 2) stably expressing either siRNA-resistant GFP-RNF168 wild type (Figure 2) or GFP-RNF168 T5 mutant lacking the PID domain (Figure 8).

Figure 2 examines recruitment of PALB2 in irradiated S/G2 cells. Recruitment requires RNF8, RNF168 and BRCA1, but not RAP80. Rad51 recruitment also requires RNF8 and RNF168. Most of these observations are not novel and needs to be indicated

Figure 3 claims that RNF168 does not affect recruitment of BRCA1 to HR sites in S/G2 cells, but there is some reduction of recruitment of BRCA1 at HR sites in cells transfected with siRNF168 (as also shown in panel C), and others have reported that BRCA1 recruitment to IRIF is dependent on RNF168. However, the text (subsection “BRCA1 recruitment to HR sites does not require RNF168”) says: "knock-down of RNF168 still allowed significant BRCA1 recruitment". The wording should be different here.

The co-precipitation experiments shown in Figure 4 are not clean enough. These experiments should show better that PALB2 binds to RNF168.

Figure 4: We see the mutant GST-PALB2 but the WT GST-PALB2 is missing.

---

## [Author Response]

*Essential revisions:*

*Since it has been previously shown that BRCA2 binds to the WD40 β-propeller domain of PALB2, the fact that RNF168 interacts with the exact same region of PALB2 (Figure 4) raises a few questions that should be discussed in more detail (subsection “A new RNF168-dependent mechanism for PALB2 recruitment”); especially in the light of the simplified model shown in Figure 8 that is lacking BRCA2 and RAD51. For instance, do RNF168 and BRCA2 interact with PALB2 in a mutually exclusive manner? Does RNF168 eventually compete with BRCA2 for PALB2 binding? How is the potential handover of PALB2 from RNF168 to BRCA2 regulated?*

*All interaction studies were performed with (transiently) overexpressed, epitope-tagged versions of RNF168. To address this issue, you could co-immunoprecipitate endogenous RNF168 from cell lysates and probe the bound fractions for PALB2 and BRCA2. If this is not possible for technical reasons, GFP-RNF168 pulldown fractions (Figure 4) should also be probed for the presence of endogenous BRCA2.*

We have attempted several co-immunoprecipitation (co-IP) experiments using commercially available RNF168 antibodies. However, the IP efficiencies proved to be (too) low and high background levels of PALB2 and RNF168 binding were experienced in these IPs despite stringent IP/wash conditions. This precluded us from examining whether endogenous RNF168 interacts with PALB2 and BRCA2. Instead, we have performed pulldown experiments using cells transiently expressing GFP-RNF168. These new pulldown experiments revealed an association between GFP-RNF168 and PALB2 (new Figure 4), corroborating our previous findings. In addition to PALB2, we also found BRCA2 to associate with GFP-RNF168 in these experiments (new Figure 4; Figure 4—figure supplement 1). To validate these findings, we performed in vitrobinding assays using purified PALB2, BRCA2 and RNF168 proteins. In these experiments, we initially used a smaller chimeric BRCA2 protein of 1009 amino acids (piccolo BRCA2 or piBRCA2), which interacts with both PALB2 and RAD51 (Buisson et al. 2010). Indeed, we found that recombinant PALB2 interacts with piBRCA2 and RNF168 (new Figure 4). Similar results were obtained when using purified full-length BRCA2 protein of 3330 amino acids (Figure 4—figure supplement 1). PALB2 interacts with both RNF168 (Figure 4) and BRCA2 (Oliver et al. 2009) through its WD40 domain. Thus, our findings suggest that RNF168 and BRCA2 interact with PALB2 through its WD40 domain in a non-mutually exclusive manner. To incorporate these new findings, we have included an extended model that also shows BRCA2 and RAD51 in Figure 8.

*It is shown that RNF168 depletion results in hypersensitivity to olaparib treatment and decreased HR efficiency (Figure 2), but not to the same extent as compared to BRCA2 depletion, which is expected. The question remaining is whether RNF168 acts epistatically to PALB2 with regards to HR or in genetically separable pathways. It is suggested to perform clonogenic PARP inhibitor survival assays using single and combined depletion of RNF168 and PALB2.*

Following the reviewers’ suggestion, we have performed clonogenic PARPi survivals using single and double knock-down of RNF168 and PALB2. While knock-down of RNF168 leads to a moderate sensitivity to PARPi, the additional knock-down of PALB2 further sensitized RNF168-depleted cells to the same levels as PALB2 single knock-down cells. These findings indicate that RNF168 is epistatic to PALB2 in HR. See subsection “BRCA1 recruitment to HR sites does not require RNF16” and new Figure 2; Figure 2—figure supplement 1.

*The PALB2-RNF168 interactions shown in Figure 8 are not convincing. The proteins are grossly overexpressed. A question remaining is whether the endogenous proteins interact.*

As indicated above, for technical reasons we have been unable to address whether endogenous PALB2 and RNF168 interact. However, the RNF168-PALB2 interaction studies in Figure 8 are in our opinion convincing, as the pulldowns show a high degree of specificity. Near equal amounts of GFP-NLS, RNF168^WT^, RNF168^Δ1-189^ (T1), RNF168^Δ191-382^ (T2), RNF168^Δ384-571^ (T3), RNF168^Δ479-571^ (T4) and RNF168^Δ525-571^ (T5) were pulled down. Proteins containing the PID domain (WT, T1 and T2), associated with PALB2 at very similar levels, while those lacking this domain (T3, T4 and T5, or GFP-NLS) failed to bind PALB2 efficiently. These findings, together with the interaction between recombinant PALB2 and RNF168 (Figure 4), and the finding that RNF168 lacking its PID domain fails to rescue PALB2 IRIF (Figure 4) and PARP inhibitor sensitivity (new Figure 8) in RNF168-depleted cells illustrate the functional significance of the PID domain in RNF168 for the PALB2-RNF168 interaction.

*To further corroborate the role of the RNF168-PALB2 interaction in HR, it would be important to examine olaparib hypersensitivity of RNF168-depleted cells (Figure 2) stably expressing either siRNA-resistant GFP-RNF168 wild type (Figure 2) or GFP-RNF168 T5 mutant lacking the PID domain (Figure 8).*

To address this issue, we used the Flp-In/T-REx system to establish HeLa clones stably expressing inducible siRNA-resistant GFP-tagged RNF168 alleles that were either able (WT) or unable (T5) to interact with PALB2. HeLa cells stably expressing GFP-NLS were generated to serve as a control. To perform PARPi survival assays, we knocked-down endogenous RNF168 with siRNAs in these stable HeLa cells and induced ectopic expression of RNF168 alleles with doxycycline. This expression was shut down by doxycycline wash-out 12-24 hours before cells were subjected to a 24 hour pulse of 1 µM PARPi (Figure 8), resulting in ectopic RNF168 expression levels in the near-physiological range during the treatment with PARPi (Figure 8; Figure 8—figure supplement 3). Using this approach, we found expression of GFP-RNF168^WT^ to almost fully rescue the PARPi sensitivity of RNF168-depleted cells, while expression of GFP-NLS or GFP-RNF168_Δ525-571_ (T5) failed to rescue this defect (Figure 8). Notably, we observed this effect in two independent clones expressing RNF168_WT_ or RNF168_Δ525-571_ (T5). These findings suggest that the interaction between PALB2 and the C-terminal PID domain of RNF168 contributes to efficient DSB repair by HR. See subsection “PALB2 recruitment requires its interaction with the C-terminus of RNF168”.

*Figure 2 examines recruitment of PALB2 in irradiated S/G2 cells. Recruitment requires RNF8, RNF168 and BRCA1, but not RAP80. Rad51 recruitment also requires RNF8 and RNF168. Most of these observations are not novel and needs to be indicated*

We would like to point out that the existing literature on the role of RNF168 in HR is quite mixed. While a number of studies have previously reported a role for RNF8 in regulating HR through different mechanisms (Huang et al. 2009; Lu et al. 2012; Zhang et al. 2012), other studies have reported that loss of RNF8 or RNF168 did not affect HR (Meerang et al. 2011; Sy et al. 2011; Munoz et al. 2012; Nakada et al. 2012). Thus, the role of RNF8 and particularly RNF168 in HR is currently unclear. More importantly, recent studies have suggested that PALB2 recruitment requires RNF8 through a mechanism that depends on RAP80 (Zhang et al. 2012). However, our work shows that RAP80 does not contribute to PALB2 recruitment (Typas et al. 2015), which is consistent with several studies showing that RAP80 does not contribute to, or may even antagonize HR (Coleman and Greenberg 2011; Hu et al. 2011; Typas et al. 2015). We have now indicated that there is no clear consensus on the role of RNF168 in HR and we cite the relevant literature in which conflicting results have been reported (see–subsection “RNF168 promotes PALB2 recruitment to DSBs and execution of HR”).

*Figure 3 claims that RNF168 does not affect recruitment of BRCA1 to HR sites in S/G2 cells, but there is some reduction of recruitment of BRCA1 at HR sites in cells transfected with siRNF168 (as also shown in panel C), and others have reported that BRCA1 recruitment to IRIF is dependent on RNF168. However, the text (subsection “BRCA1 recruitment to HR sites does not require RNF168”) says: "knock-down of RNF168 still allowed significant BRCA1 recruitment". The wording should be different here.*

We now state that: “knock-down of RNF168 still allowed BRCA1 recruitment in cells that failed to form 53BP1 foci (Figure 3)” (see subsection “BRCA1 recruitment to HR sites does not require RNF168”). We also explain in the same section that the knock-down of RNF168 shifted the average BRCA1 foci size from that of the typical large signalling foci (0.16 µm^2^) to that in the range observed for HR proteins (0.08 µm^2^; Figure 3). In other words, we lose the larger RAP80-BRCA1 foci in RNF168-depleted cells, but we retain the smaller HR- associated BRCA1 foci. We also state that: “Although RNF168-depleted cells do form IR-induced BRCA1 foci in S/G2, we noted that the absolute number of BRCA1 foci is decreased compared to control S/G2 cells (Figure 3). […] In summary, our findings suggest that while RNF168 loss affects the formation of large BRCA1 foci, which arise from RAP80-dependent BRCA1 recruitment during DSB signaling, it does not impair the formation of small BRCA1 foci in S/G2 cells that are typical for factors involved in HR.”

*The co-precipitation experiments shown in Figure 4 are not clean enough. These experiments should show better that PALB2 binds to RNF168.*

We have repeated the co-immunoprecipitation experiments and provide clearer images that show the interaction between RNF168 and PALB2 (see new Figure 4, Figure 4, and Figure 4—figure supplement 1).

*Figure 4: We see the mutant GST-PALB2 but the WT GST-PALB2 is missing.*

GST-PALB2-WT is shown in Figure 4and interacts with RNF168-His. The deletion fragments T1 until T5 are shown in Figure 4.